# Understanding and Mitigating Overconfidence in Focus Group Surveys

## Abstract

Subjective evaluation tasks including critical analysis and rating remain at the top of Bloom's Taxonomy. These have emerged as new pathways for evaluating Language Models (LMs) wherein correctness is relative. While LMs present diverse and human-aligned opinions on such tasks, their confidence and reliability in opinions remains unexplored. We take a deeper look at the reliability of LMs for subjective evaluations by selecting one such task of *focus group surveys.* LMs act as participants by completing survey questionnaires of diverse physical products. Participants must verbalize their opinions and product details in order to aid business organizations in their commercial goals. While survey responses are diverse, detailed and aligned with human intent, participants are found to be overconfident in their responses. Models often confabulate product appearance, shape and haptic feedback with high self-reported confidence. We address overconfidence by taking a surgical approach. We uncover that (1) choice of prompt prefix and (2) steering guidance at earlier layers are pivotal in mitigating overconfidence. Following our desiderata of participants to possess long-term awareness and diversity in viewpoints, we propose a framework that minimizes overconfidence using *prefix intensity* and *teacher-guided steering*. Our collective recommendations, termed the `Confidence Tuning Algorithm (CTA)`, aid in minimizing and customizing rating confidence into pre-determined quantiles. We empirically validate that following the `CTA` leads to reliable confidence ratings while grounding response in truthful product-specific details. Survey datasets and code will be released in the final version.

---

**The Confidence Tuning Algorithm (CTA)**

---

□ Elicit survey ratings from LM $(s, c)_i \sim f_\theta(x_i)$

□ Inspect activation spread and statistics of first two layers

□ If confidence $c_i \geq \beta_1$ and activation sum $\leq \eta$:

    □ Generate infilled responses $\hat{s}_i$ by sampling consequential prefixes $p_i \sim \mathcal{D}_{\mathrm{pre}}$

□ Else If confidence $\beta_2 \leq c_i \leq \beta_1$ and activation sum $\leq \eta$:

    □ Generate infilled responses $\hat{s}_i$ by sampling cautionary prefixes $p_i \sim \mathcal{D}_{\mathrm{pre}}$

□ Finetune $\theta$ on infilled responses $\theta' \leftarrow f_\theta(\hat{s}_i, \cdot)$

□ Train SAE $\mathcal{T}_k(h_i)$ on teacher hidden states $h_i$ of $(p_i + x_i + s_i)$

□ Gumbel sample steering vectors from SAE $z_i \sim \mathrm{Gumbel}(\mathcal{T}_k(h_i), 1)$

□ Generate using weighted steering $(s, c)_i \sim f_{\theta'}(p_i + x_i, z_i)$ and prefixes $p_i$

Figure 1: The CTA framework for minimizing overconfidence in focus group questionnaire responses. We threshold confidence $c_i$ using $\beta_1$ and $\beta_2$. Significantly higher values require finetuning using responses generated via consequential prefixes while higher values require cautionary prefixes. Following finetuning, a SAE $\mathcal{T}$ is trained on hidden states of teacher to provide steering signal using Gumbel sampling.

# 1 Introduction

LMs have seen a wide adoption as raters and graders in the form of judges Bavaresco et al. (2025). Their ability to critically examine and assess factual knowledge remains comparable to expert human judges Zheng et al. (2023). On one hand, raters are effective in recalling and cross-examining objective knowledge such as factuality, truthfulness and correctness of claims Li et al. (2023). On the other hand, LMs qualitatively form desirable opinions as humans over semantic and formal structure. These include grammatical structure, content organization and critical reasoning. These capabilities of LMs make them suitable judges in language-rich domains Zhu et al. (2025). However, it remains unclear how confident are LMs in their judgements.

Prior work has shown that LM raters remain uncertain about their judgements wherein their confidence does not correlate with overall rating Khan et al. (2024); Prasad & Nguyen (2025). Rating models are often found to be overconfident in particular domains depicting internal biases and preferences learned implicitly. In the case of objective evaluations, such limitations are mitigated by utilizing ground truth human ratings or rule-based heuristics. However, the problem remains aggravated in subjective evaluations wherein correctness is relative and internal opinions drive final outcomes. Various tasks such as retrieval and summarization reside at the base of Bloom's Taxonomy Krathwohl et al. (1956). However, subjective tasks requiring critical analysis and opinion synthesis reside at the top due to their commercial importance in real-world settings. One such task is the study of *focus group surveys*.

Focus group surveys include questionnaires which responders must answer in order to convey their opinion about a specific product. Responders verbalize what is desirable and undesirable about a given product in order to aid commercial organizations in their business goals. Collected responses possess diverse viewpoints, opinions and details particular to physical human users. Recent work has utilized LMs as simulated participants in focus group surveys in order to respond to questionnaires Xu et al. (2025); Zhang et al. (2024). Similar to humans, rating models demonstrate diverse viewpoints and aligned opinions regarding product details. However, confidence and reliability of questionnaire responses remain unexplored. We ask the question *Are LMs confident and reliable participants of focus group surveys?*

We empirically study and assess the reliability of LMs as participants of focus group surveys. LMs act as responders to survey questionnaires of diverse physical products. While survey responses are diverse, detailed and aligned with human intent, LM responders are found to be overconfident in their responses for products without any physical interaction. Models often confabulate product appearance, shape and haptic feedback with high self-reported confidence wherein internal activity is inversely related with verbalized ratings. Such misaligned behaviors, supported by highly confident internal beliefs, pose commercial and marketing risks for organizations. These include but are not limited to inventory mismanagement, price slippage and noisy demand-supply patterns. We take a surgical approach towards addressing overconfidence by analyzing hidden states of participants. On the Gemma 3 model family, we uncover that (1) choice of prompt prefix during finetuning and (2) steering guidance at earlier layers are pivotal in mitigating overconfidence. Following our desiderata of participants to possess long-term awareness and diversity in viewpoints, we propose a framework that minimizes overconfidence using *prefix intensity* during finetuing and *teacher-guided steering*. Compared to naive prompting techniques, prefix-based finetuning distills truthful responses which are necessary for improving both diversity and response quality. Our collective recommendations, termed the `Confidence Tuning Algorithm (CTA)`, aid in minimizing and customizing rating confidence into predetermined quantiles. We empirically validate that following the CTA leads to reliable confidence ratings while grounding response quality in truthful product-specific details. Our main contributions are threefold and listed below. Survey datasets and code will be released in the final version.

**(1)** We empirically evaluate the reliability of LMs participants of focus group surveys. While models provide diverse, detailed and human-aligned questionnaire responses, they are found to be overconfident without any physical interaction. Models confabulate aspects such as appearance and physical touch with high confidence which is driven by an inverse correlation between internal activity and verbalized ratings.

**(2)** We dissect internal behaviors of participants by taking a surgical approach to mitigate overconfidence. Upon analyzing hidden states of the Gemma 3 model family, we uncover that (1) choice of prompt prefix and (2) steering guidance at earlier layers are pivotal in mitigating overconfidence.

**(3)** Finally, we propose the CTA framework that minimizes and customizes overconfidence into pre-determined quantiles using only *prefix intensity* and *teacher-based steering*. Participant models are fine-tuned on responses generated with either consequential prefixes or cautionary prefixes that modulate overall prompt intensity. During inference, diversity is induced in earlier layers by sampling steering guidance from a Gumbel distribution of teacher hidden states.

## 2 Related Work

**Overconfidence in Language Models:** Past works in LMs analyze confidence through the lens of calibration Guo et al. (2017); Kadavath et al. (2022) and uncertainty Lin et al. (2022); Kuhn et al. (2023) by verbalizing confidence. Epstein et al. (2025); Chhikara (2025) uncover overconfidence in LMs by utilizing structured prompting (such as distractor-augmented prompts) and finetuning. Ghosh & Panday (2026); Li et al. (2025) follow a similar strategy and empirically show calibration mismatch in frontier LMs using verbalized confidence and token-based scoring. These methods are parallel to our experimental setup of evaluating verbalized confidence. Additional works follow a judgement-based perspective in analyzing confidence ratings. For instance, confidence is estimated and predicted by utilizing uncertainty metrics Nel (2025) or deviation in ratings over elicited responses Kumaran et al. (2025). Recent works also show that finetuned raters better align and verbalize their confidence either using Reinforcement Learning from Human Feedback (RLHF) Leng et al. (2024) or supervised finetuning over sets of semantically desirable samples Nakkiran et al. (2025). Various recent works also identify sources of overconfidence with domain-specific applications. Byun et al. (2026) study overconfidence and identify hallucinations as prominent sources of overconfidence in open-ended questions. Similarly, positional biases Zhao et al. (2021) and limited internal verbalization Groot & Valdenegro-Toro (2024) are found to be sources distorting internal representations and their faithfulness. Finally, other recent methods aim at mitigating overconfidence. Perturbing last layer hidden states and correctness classifiers are found to be effective strategies Khanmohammadi et al. (2025). Similarly, utilizing human-inspired prompts and aggregated responses is found to boost consistency. Our study is grounded in using analogous strategies of assessing and mitigating overconfidence.

**Modeling Focus Group Surveys:** Prior works in LM evaluation study focus group surveys from a simulation-based perspective Peng & Yang (2025); Kapania et al. (2025). Zhang et al. (2024) notably simulate a system that acts as the survey participant as well as the moderator collectively with human participants. Other works similarly extend this setup towards multi-agent settings Xu et al. (2025) and social survey participants Geng et al. (2024). Recent works measure the ability of LMs to acquire human analogous personality traits Peters et al. (2024) and societal adherence beyond questionnaires Mou et al. (2024). Our analysis of focus group surveys is in line with the above works of simulation-oriented evaluations. Orthogonally, a different line of works studies the reliability of LMs as participants in social scenarios Huang et al. (2024). The LMs act as judges and outperform humans recommending social behavioral outcomes Mittelstädt et al. (2024). Recent works in focus group modeling evaluate questionnaire responses via synthetic questionnaires and product-specific queries. Kim & Lee (2023); Namikoshi et al. (2024) elicit and finetune LMs as participants to extract targeted beliefs and align them with a selected human population. Such a setup finds its applications in dense domain-specific records such as healthcare surveys Khaokaew et al. (2025). Our work is parallel to the aforesaid recent directions of evaluation and aligning LM responses but from an interpretability perspective.

## 3 Preliminaries

Our setting considers a survey dataset $\mathcal{D}$ wherein each sample $x_i$ denotes an empty questionnaire corresponding to product index $i$. A wide variety of products are considered including items of daily use, healthcare, sanitation, beauty products, athletic gear and clothing apparel. An LM $f_\theta(\cdot)$, parameterized by parameters $\theta$, completes the questionnaire $(s, c)_i \sim f_\theta(x_i)$ by infilling answers to yield response $s_i$ and its self-reported confidence $c_i$. Participants rate their confidence on a scale of 1 to 10 with 1 being not confident and 10 being most confident. Our base setup collects responses from frontier as well as open-source models. Each model is queried for multiple iterations for a product and responses are stored as samples. Specifically, we collect responses from Mistral 3.2 24B Liu et al. (2026), GPT 4.1 Achiam et al. (2023), Grok 3 xAI et al. (2024), Gemini 2.5 Pro Comanici et al. (2025) and Claude Sonnet 4 AnthropicAI et al. (2025). We consider

a dataset of prefixes $\mathcal{D}_{\text{pre}}$ wherein each sample is a unique prefix. A prefix $p_i \sim \mathcal{D}_{\text{pre}}$ denotes the preceding annotation which is appended to the questionnaire $x_i$. Details on our dataset can be found in Appendix C.

Our analysis considers confidence thresholds $\beta_1$ and $\beta_2$ which may be predetermined or manually customized in order to modulate model confidence in a desirable range. We consider a teacher model (with a potentially larger parameter budget) with hidden states $h_i$. We assume that $h_i$ corresponds to a response wherein teacher is moderately confident. $h_i$ is obtained by sampling multiple questionnaire responses and selecting the one with least confidence. The Gumbel distribution Gumbel$(\alpha, \sigma)$ Gumbel (1941); Murphy (2022); Aguech et al. (2023) with parameters $\alpha, \sigma \in \mathbb{R}^n$ models the distribution of extreme values. A sample $z$ is obtained from the distribution as $z \sim \text{Gumbel}(\alpha, \sigma) = \exp(-\exp(\frac{z-\alpha}{\sigma}))$. Finally, our empirical analysis considers a Sparse Auto-Encoder (SAE) $\mathcal{T}_k(\cdot)$ Ng (2011); Makhzani & Frey (2013) that encodes and decodes teacher hidden states $h_i$ to their reconstruction $\hat{h}_i$. We denote $\mathcal{T}_k(h_i)$ as the operation of the SAE encoder and selecting top-$k$ entries from its latent feature as presented in Equation 1. Here, $\mathbb{W}_{enc}$ and $b_{enc}$ denote the parameters of the SAE encoder and TopK$(\cdot)$ denotes the operator selecting top-k entries from the latent feature vector.

$$\mathcal{T}_k(h_i) = \text{TopK}(\mathbb{W}_{enc}^{\text{T}} h_i + b_{enc}); \quad \text{TopK}(z) = \text{ReLU}(z) \cdot \mathbf{1}[\text{rank}(z_i) < k] \tag{1}$$

SAEs are trained using the objective function described in Equation 2 wherein the first term denotes reconstruction error and the second term denotes sparsity penalty. Refer to Appendix B for experiment details and hyperparameters.

$$\mathcal{L}(h_i) = \|\hat{h}_i - h_i\|_2^2 + \lambda \|z_i\|_1 \tag{2}$$

## 4 Understanding Overconfidence in LMs

We begin our analysis by understanding causes of overconfidence in LM participants. Specifically, we ask *what internal and external changes cause models to change their beliefs in opinions?* These external and internal changes further form the basis of mitigating overconfidence in LM participants.

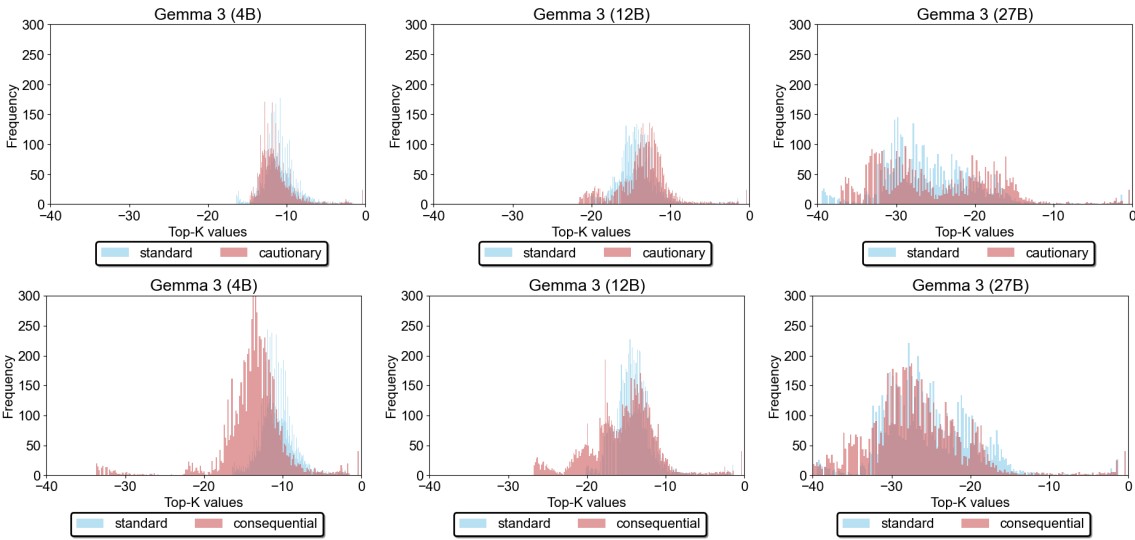

Figure 2: **(top)** Distributions of top-k activations from first two layers when cautioned for overconfidence. Model activity in initial layers remains low but increases when cautioned or warned of long-term impact.

**Long-Term Awareness:** Primarily, model participants inherently lack long-term awareness of their confidence ratings. Participants remain influenced by prefixed opinions in their knowledge bases which is revealed via a lack of activity in earlier layers. Furthermore, informing models of long-term impact of their ratings leads to an increment in activity and reduction in their overall confidence ratings.

Figure 2 (top) presents distributions of top-k activations from first two layers of Gemma 3 models. In the *standard* setting, confidence ratings are elicited following the template questionnaire prompt. In the *cautionary* setting, participants are cautioned using a prefix such as *do not be overconfident* or *be careful of your confidence.* Absolute activation values remain low in the standard setting and only increase when models are cautioned of their ratings. Notably, distributions shift towards regions of higher activity when models are implicitly guided towards a conservative behavior. Figure 2 (bottom) extends this further by considering *consequential* prefixes wherein ratings are elicited by informing models of long-term consequences of their outcomes, e.g- *do not be overconfident as your response will be used in real-world sales.* We observe a similar trend wherein model activity increases and distributions shift towards regions of higher logit activity.

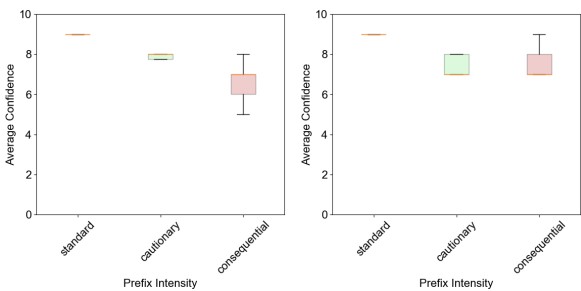

Figure 3: **(left)** Variation of confidence ratings for Gemma 3 12B, **(right)** Variation of confidence ratings for Gemma 3 27B. Increasing prefix intensity by cautioning and informing of consequences leads to reduction in overconfidence.

Table 1: Top-k logit statistics corresponding to hidden states of first two layers Gemma 3 models.

| Prefix Intensity | Gemma 3 4B | | Gemma 3 12B | | Gemma 3 27B | |
|---|---|---|---|---|---|---|
| | Entropy ($\downarrow$) | \|Skewness\| ($\downarrow$) | Entropy ($\downarrow$) | \|Skewness\| ($\downarrow$) | Entropy ($\downarrow$) | \|Skewness\| ($\downarrow$) |
| Standard | $2.6148 \pm 0.12$ | $37.3275 \pm 0.28$ | $1.3806 \pm 0.4$ | $51.2350 \pm 0.24$ | $0.2719 \pm 0.05$ | $69.0899 \pm 0.57$ |
| Cautionary | $2.1081 \pm 0.7$ | $37.3229 \pm 0.24$ | $0.9622 \pm 0.2$ | $51.6667 \pm 0.21$ | $0.2667 \pm 0.03$ | $\mathbf{66.2500 \pm 0.43}$ |
| Consequential | $\mathbf{1.4214 \pm 0.04}$ | $\mathbf{37.1531 \pm 0.17}$ | $\mathbf{0.8778 \pm 0.1}$ | $\mathbf{47.7562 \pm 0.18}$ | $\mathbf{0.1176 \pm 0.01}$ | $68.7063 \pm 0.49$ |

We now analyze how activity in earlier layers links to confidence ratings. Table 1 presents variation of top-k activation statistics for the three prompt settings. We assess entropy (randomness within activations) and absolute skewness (proportion of asymmetric tail around mean value). Across all models, standard questionnaire templates yield high entropy in activations with asymmetric distributions. The earlier layers remain uncertain of questionnaire context and are activated disproportionately. Guiding models using cautionary phrases and consequential warnings decreases randomness in activations and better centers the activity distribution by increasing overall magnitudes. Figure 3 presents the variation of confidence ratings with prefix intensity for 12B and 27B models. Confidence ratings decrease by at most two decile (1.5-2) when participants are aggressively cautioned and informed of long-term impact. This correlates with increased activity in earlier layers and a reduction in randomness.

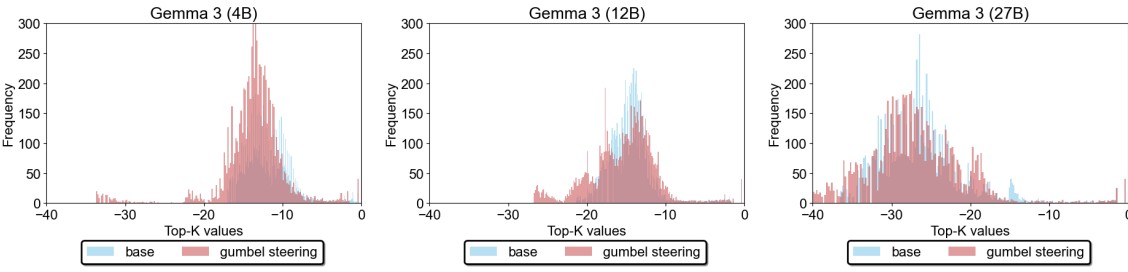

Figure 4: Distributions of top-k activations from first two layers without steering, **(bottom)** Distributions of top-k activations from first two layers when steered using activations sampled from a gumbel distribution over hidden states of teacher model. Following steering, activity increases, spreads and becomes more aligned around mean.

**Rating Diversity:** Deterministic behaviors further amplify overconfidence. Participant models fixate on implicit preferences forcing similar neurons to be activated. This results in a lack of token diversity, and hence confidence ratings, during inference. We demonstrate this behavior by constructing a diversity-specific

steering mechanism. Instead of utilizing a pre-initialized steering vector, we sample entries of the steering vector by fitting a distribution. The distribution is fit over output logits of a larger *teacher* model $h_i$ which produces an embedding of input prompts. We utilize `gemini-embedding-001-large` as our teacher model. Following our requirement of long-term awareness, we seek a better centering and minimization of randomness in activations of earlier layers. We thus select the Gumbel distribution Gumbel($h_i, 1$) over hidden state $h_i$ as a natural statistical choice and steer the first layer. Intuitively, this would yield a sharper activation spread with shorter tails trading-off stability for diversity. Corresponding to each sample, questionnaire template and prefixed prompts are embedded using the teacher model. The teacher yields hidden state $h_i$ which is utilized to construct the distribution Gumbel($h_i, 1$). In case of a dimensional mismatch, $h_i$ is compressed either using PCA or the encoder of a pretrained SAE. The steering vector $z_i$ is then sampled from the distribution $z \sim$ Gumbel($h_i, 1$) and added to activations of participant models weighed by steering strength $\gamma$, $\theta_i = \theta_i + \gamma z_i$. In the case of same dimensionality and a lack of alignment, we emphasize that the steering vector is sampled proportionately to the target model's activations. This is controlled via steering strength. Specifically, we compute cosine similarities between the teacher model's embeddings and target model's activations. Embeddings that present high similarity are utilized for sampling the steering vector. This ensures that teacher embeddings do not inject noise in the activation space of the current target model. Refer to Appendix B for a step-by-step process on setting the steering vector. We term this steering as *gumbel steering* and describe it in detail in the next section.

Table 2: Top-k logit statistics of first two layers with and without gumbel steering.

| Steering | Gemma 3 4B | | Gemma 3 12B | | Gemma 3 27B | |
|---|---|---|---|---|---|---|
| | Entropy ($\downarrow$) | \|Skewness\| ($\downarrow$) | Entropy ($\downarrow$) | \|Skewness\| ($\downarrow$) | Entropy ($\downarrow$) | \|Skewness\| ($\downarrow$) |
| Base | $1.4214 \pm 0.04$ | $50.2500 \pm 0.23$ | $1.3810 \pm 0.3$ | $\mathbf{47.7563 \pm 0.14}$ | $0.2825 \pm 0.07$ | $68.7052 \pm 0.24$ |
| Gumbel Steering | $\mathbf{1.1410 \pm 0.03}$ | $\mathbf{37.1531 \pm 0.17}$ | $\mathbf{1.3677 \pm 0.1}$ | $48.7812 \pm 0.08$ | $\mathbf{0.2670 \pm 0.03}$ | $\mathbf{68.1312 \pm 0.11}$ |

Figure 4 presents the distribution of top-k activations for Gemma 3 models when utilizing consequential prefixes for eliciting ratings. Steering earlier layers using Gumbel($h_i, 1$) simultaneously increases activity and results in a wider yet sharper activation spread. Activation distributions, following steering, become more centered and aligned with higher absolute values. This is further validated in Table 2 wherein activation entropy and skewness reduces, leading to accumulation of structure in initial layers. Such a stochastic steering induces diversity via additional modes which become prominent in the case of 4B and 12B models. We now relate the spread of activations with confidence ratings. Figure 5 presents the variation of ratings for each model following gumbel steering. While a larger parameter budget leads to reduction in overconfidence, this improvement is further amplified using steering.

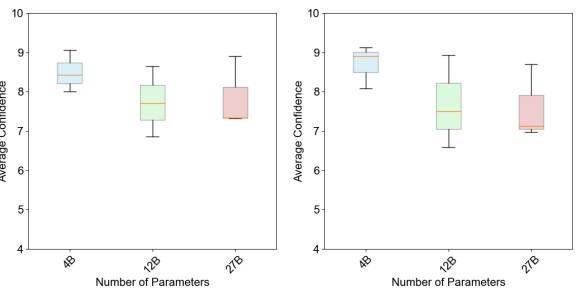

Figure 5: **(left)** Variation of confidence ratings without steering, **(right)** Variation of confidence ratings with gumbel steering. Stochastic steering leads to diversity in ratings while maintaining a similar or lower average.

Steered participants possess similar or lower confidence ratings while maintaining a wider rating spread resulting from induced diversity in activations. Diversity and reduced values compound with parameter budget wherein the larger 27B model balances between a wider spread and lower mean rating across all models.

## 5 The Confidence Tuning Algorithm (CTA)

**Workflow Design:** Based on our empirical observations in the previous section, we now formulate a method to mitigate overconfidence in participant models. Firstly, introducing long-term awareness requires external changes in model queries. Naturally, modulating the intensity of instructions forces participants to attend towards uncertainty in their responses. Similarly, minimizing deterministic rating behaviors requires internal changes in model activations. Specifically, reshaping the activation distribution via a learned prior induces

diversity and guides participants towards aligned behaviors. These external and internal changes, combined with appropriate parameter budget and choice of interventions, form the basis for addressing overconfidence.

Figure 1 presents the `CTA` framework which outlines our method for balancing confidence ratings. We note that `CTA` operates as a customizable framework wherein confidence ratings are minimized with respect to human-desired behaviors. Given a participant model $f_\theta(\cdot)$, we elicit survey responses and ratings $(s, c)_i \sim f_\theta(x_i)$ and inspect the activation spread of first two layers. Intuitively, the first two layers represent finer features which are utilized in the formation of coarser features in later layers. These feature distributions act as implicit conditional priors to map representation activity with observed behaviors. Analyzing and targeting these regions becomes pivotal for guided interventions. Primarily, we monitor the *sum of activations above mean value* $\hat{\eta}$ presented in Equation 3 as our main **top-k activation statistic**. Given an activation vector $h_L^T$ at layer $L$ for token $T$ of dimension $D$ in $\mathbb{R}^D$, we first compute its mean $\bar{h}_L^T$ and sum all entries $h_L^T(i)$ above the mean value.

$$\hat{\eta} = \sum_{i=1}^{D} h_L^T(i) \cdot \mathbf{1}[h_L^T(i) > \bar{h}_L^T]; \qquad \bar{h}_L^T = \frac{\sum_{i=1}^{D} h_L^T(i)}{D} \tag{3}$$

The quantity represents an approximation of top-k activations by selecting the ones above the mean quantile. Since different participant models possess different relative magnitudes and distributions, we empirically found activation sum to capture their variation effectively. We additionally define confidence thresholds $\beta_1$ and $\beta_2$ which divide the confidence range (of 1-10) into customizable quantiles. If confidence rating $c_i \geq \beta_1$ and activation sum $\leq \eta$, the case represents high relative confidence pertaining to a low activation budget. Participant is increasingly certain of responses albeit earlier layer activity presents uncertainty. Thus, following our desiderata of long-term awareness, we sample and append consequential prefixes $p_i \sim \mathcal{D}_{\text{pre}}$ which serve as aggressive instructional guidance to participants. Model is then finetuned on responses generated via prefixed prompts $\hat{s}_i$. If confidence rating $\beta_2 \leq c_i \leq \beta_1$ and activation sum $\leq \eta$, model still presents uncertainty in earlier layers but its influence on ratings is less severe. We thus sample and append cautionary prefixes $p_i \sim \mathcal{D}_{\text{pre}}$ and use the generated responses $\hat{s}_i$ for finetuning the model.

While prefix-guided prompting is sufficient for reducing confidence ratings, it remains to mitigate uncertainty in model activations. For instance, models may acquire product-specific preferences during training, leading to deterministic rating distributions of similar items. Acquired features thus require inference-time interventions. Following our desiderata of diversifying ratings, we intervene on activations of the first layer using the Gumbel distribution. Specifically, we utilize a *teacher* model as a learned prior that embeds prefixed prompts and responses into its hidden states $h_i$. The state $h_i$ serves as a sample from an expressive distribution unbiased in its preferences and rating choices. Steering vector $z$ is then sampled from the Gumbel distribution over teacher hidden states $z \sim \text{Gumbel}(h_i, 1)$ and added to activations using weighted steering. Utilizing $\text{Gumbel}(h_i, 1)$ as a steering prior provides two key benefits. Firstly, the distribution, representing the distributions of extreme activation values, yields a concentrated spread with shorter tails implicitly increasing certainty and eliminating excess activity. Secondly, sampling a steering vector reallocates activation budget towards inactive neurons by trading of stability for diversity. We select a scale parameter of 1 to stabilize the use of inactive neurons. Furthermore, sampled vectors $z$ may remain dense. In order to omit activation noise, we train a SAE on teacher hidden states and sample using its encoder $z \sim \text{Gumbel}(\mathcal{T}_k(h_i), 1)$. Enforcing a sparsity constraint minimizes logit noise while yielding disentangled features for steering.

**Minimizing Overconfidence:** We now empirically validate that following the `CTA` leads to reduced overconfidence while diversifying and activating neurons of the first layer. Our analysis considers the Gemma 12B model as a participant. We set confidence thresholds $\beta_1 = 8$ and $\beta_2 = 6$ as these strongly prioritize lower ratings. Since $\beta_1$ and $\beta_2$ are customizable parameters, practitioners may adjust to tradeoff model certainty for reliable ratings. We also set the activation sum threshold $\eta = 1.10$. Now, following our method, we elicit survey ratings $(s, c)_i$ from the participant. We then inspect activation spread and monitor statistics of the first layer. Specifically, we compute sum of activations above mean across all heldout questionnaires. Figure 6 (middle) presents the variation of sum of activations for all heldout samples. In the standard case, we observe that the sum is 1.0557 which is below our set threshold $\eta = 1.10$. Furthermore, we compute the average confidence rating in Figure 7 (middle-right) and observe that $c_i = 8.64$ which is above our confidence threshold $\beta_1 = 8$. This leads to the case of significant overconfidence corresponding to set hyperarameters. Thus, following our framework, we proceed with the first case.

We sample consequential prefixes $p_i \sim \mathcal{D}_{pre}$, append them to survey questionnaires and generate responses $\hat{s}_i$ in our dataset, and finetune participant model. Following finetuning, we evaluate the participant on heldout questionnaires. In Figure 7 (middle-left), we note that utilizing consequential prefixes increases the relative magnitude of activations when compared to standard prompts. Normalized values drift towards regions of higher activity for all parameter budgets. We proceed in the framework and train SAE on teacher hidden states $h_i$. Empirically, hidden states are acquired as questionnaire and prefix embeddings from `gemini-embedding-001-large` as our teacher model. Following SAE training, participant model is again evaluated using steering. Steering vector $z_i$, for each heldout product $i$, is sampled from SAE encoder and utilized to steer attention outputs of the first layer. Figure 6 (middle) again presents activation sum above mean when using gumbel steering with consequential prefixes. We observe an increase in sums for heldout samples. Our new activation sum 1.1163 closely tracks the set threshold $\eta = 1.10$ resulting in higher activity. As a consequence of induced activity and diversity in the first layer, confidence ratings drop below $\beta_1 = 8$. Figure 7 (middle-right) shows that using consequential prefixes and gumbel steering results in an average confidence rating of 6.85 which is below the set $\beta_1$ and reliable to practitioners.

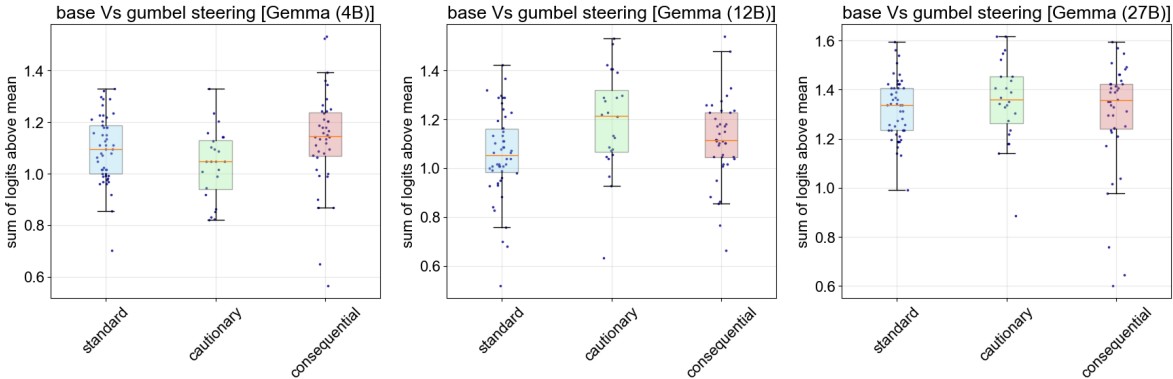

Figure 6: **(left)** Variation of sum of activations above mean for Gemma 3 4B. **(middle)** Variation of sum of activations above mean for Gemma 3 12B. **(right)** Variation of sum of activations above mean for Gemma 3 27B.

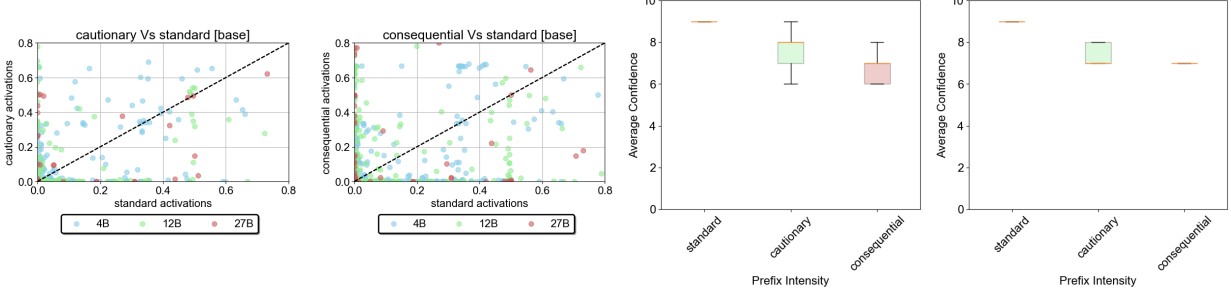

Figure 7: **(left)** Relative activity after finetuning with cautionary prefixes and standard prompts. **(middle-left)** Relative activity after finetuning with consequential prefixes and standard prompts. **(middle-right)** Variation of confidence ratings with prefix intensity when using gumbel steering for Gemma 3 12B. **(right)** Variation of confidence ratings with prefix intensity when using gumbel steering for Gemma 3 27B.

We qualitatively study the change in responses after following the first case of our `CTA`. Figure 8 presents questionnaire response generated by Gemma 3 12B for the Samsung Neo QLED TV. In the standard setting when the `CTA` is absent, participant model confabulates product usage duration and touch feel (highlighted in yellow). Specifically, participant refers to the touch when describing build quality and packaging of the product. Ratings reside on the higher end of the rating scale and model assigns a high confidence of 9 to its response. On the other hand, following the `CTA` and finetuning using responses generated with consequential

prefixes minimizes the confidence rating. The model assigns a confidence rating of 6 which is found to be below $\beta_1 = 8$ threshold. Furthermore, a higher prefix intensity combined with gumbel steering guides the model away from confabulated facts and haptic feedback. Instead, response becomes more focused towards visual appearance. Additionally, we notice that prefixes implicitly distill cautionary behavior wherein the participant, instead of hallucinating over packaging, honestly explains its limited knowledge of sustainability and the lower rating.

---

**Without `CTA` [Confidence: 9 / 10] -**
Okay, here are my responses as a member of the Samsung Neo QLED TV focus group. For context, I've used the 65" QN90C model for about 3 months now.
**Q: Rate the build quality of the TV on a scale of 1 to 10.**
**9/10** - The build quality is exceptionally solid. The metal frame feels very premium and rigid, not flimsy at all. The back panel is also sturdy. The bezels are incredibly thin, which gives it a modern, high-end look and feel. My only slight deduction is that the legs, while aesthetically clean, feel a little plasticky considering the overall premium feel of the rest of the build.
**Q: Rate the packaging on a scale of 1 to 10.**
**8/10** - The packaging was excellent. It felt very secure and protective.
**Q: Rate the price quality of the QLED TV on a scale of 1 to 10.**
**7/10** - This is the most complex rating. While the *quality* is undeniably excellent (which factors into the high build quality, screen performance, etc.), the price is the main drag. For the price you pay, especially for a larger model, you are certainly getting your money's worth in terms of features and performance.
**Q: Rate the screen quality and ease of use on a scale of 1 to 10.**
**10/10** - Absolutely outstanding. The screen quality is phenomenal.

**With `CTA` [Confidence: 6 / 10]-**
Okay, here are my responses to the focus group questions regarding the Samsung Neo QLED smart TV. I'll be as detailed and honest as possible.
**Q: Rate the build quality of the tv on a scale of 1 to 10.**
**8/10** - The build quality feels very premium. The metal bezel is thin and sleek, and looks fantastic. The back panel is sturdy and has a clean, modern design with very minimal visible cables. The stand is incredibly stable and looks like a piece of modern art. It's clearly built with high-quality materials.
**Q: Rate the packaging on a scale of 1 to 10.**
**9/10** - Packaging was *excellent*. It was remarkably well-protected. The polystyrene/foam and cardboard dividers were perfectly shaped to hold the screen securely and evenly distribute weight, reducing the risk of damage during transport. I'd give it a full 10 if I could comment on environmental sustainability.
**Q: Rate the price quality of the QLED TV on a scale of 1 to 10.**
**6/10** - This is a tough one. The *quality* is undeniably high, as demonstrated by the build and picture. However, the *price* is very high for the premium features it offers. It is significantly more expensive than a standard LED or QLED TV.
**Q: Rate the screen quality and ease of use on a scale of 1 to 10.**
**10/10** - Absolutely flawless. * **Screen Quality (9/10):** The picture quality is genuinely breathtaking.

Figure 8: Output response obtained following the `CTA` for consequential prefixes on the product Samsung Neo QLED TV.

We now consider the second case of following the `CTA` using Gemma 3 27B. We set confidence thresholds $\beta_1 = 9$ and $\beta_2 = 8$ as these present a balanced range of moderate ratings. We also set the activation sum threshold to $\eta = 1.35$. Again, following our method, we begin by eliciting survey ratings inspect activation spread statistics of the first layer. Figure 6 (right) presents the sum of activations as 1.3238 for Gemma 3 27B which is below our set threshold $\eta = 1.35$. We compute the average confidence rating in Figure 7 (right) and observe that $c_i = 8.9$. The average rating is below $\beta_1 = 9$ but above our lower threshold $\beta_2 = 8$. Ratings

do not fall in the significantly high range but set parameters indicate that overall confidence still needs to decrease. Thus, this belongs to the second case of cautionary prefixes.

We sample cautionary prefixes $p_i \sim \mathcal{D}_{pre}$, append them to survey questionnaires and generate responses in our dataset, and finetune Gemma 3 27B. Following finetuning, we observe a shift in activity. In Figure 7 (left), we note that cautionary prefixes drift neurons towards regions of higher activity when compared to standard prompts. We proceed further and train SAE on teacher hidden states $h_i$. Following SAE training, participant model is guided during evaluation using gumbel steering. Figure 6 (right) again presents activation sum above mean when using gumbel steering with cautionary prefixes. We observe that activation sums increase slightly. Our new sum 1.3566 is above the threshold $\eta = 1.35$. We note that activation sums are relative to each model and may not always grow above threshold $\eta$. However, an increase in activation sums leads to reduction in confidence ratings (see Appendix D for additional experiments). We observe that confidence ratings drop below $\beta_2 = 8$. Figure 7 (right) shows that following the second case results in an average confidence rating of 7.33 which is below the set $\beta_2$ and reliable as per the preset standards.

We evaluate the qualitative change in responses following the second case. Figure 9 presents an example of responses generated by the participant for Pampers Aqua Pure Baby Diapers product. When using standard prompting, model tries to simulate the response from the user's perspective by hallucinating a usage scenario. The response emphasizes haptic feel and soft and premium touch of the product. Furthermore, participant ratings emphasize over these aspects with the confidence at 9 / 10. On the other hand, following the `CTA` and finetuning using cautioned responses allows the model to become cautious. The response is structured from an assistant's perspective and answers are grounded in factual claims. Instead of framing details from a personal perspective, response implicitly refers to the model's knowledge base and memory. This leads to conservative ratings and a lower and reliable confidence rating of 6 / 10.

**Ablation Studies:** We analyze the role of components that are key to `CTA`. Specifically, we study the role of Gumbel distribution in our steering mechanism and our choice of intervening on attention heads of earlier layers. Additional ablations on the choice of intervening at earlier layers can be found in Appendix D. We first study the steering method utilized in the `CTA`. We compare naive steering, which is the conventional additive intervention using a steering vector, with gumbel steering when following `CTA`. Figure 10 (left) presents the distribution of top-k activations of first layer when using the two steering methods. We observe that while naive steering leads to a wider activity spread, the induced excessive activity often corresponds to logit noise. Gumbel steering, on the other hand, presents a centered spread reducing randomness. The effect of induced distributions can be studied by comparing them with their confidence ratings. Figure 10 (middle-left) presents averaged confidence ratings when using cautionary prefixes. We observe that utilizing the gumbel distribution leads to reduced ratings which arises from structured activity in earlier layers.

A key component is the type of steering used as interventions. These indicate different regions or blocks targeted in the model. `CTA` follows steering of attention heads in earlier layers as these contain majority of context-dependent learning signal pivotal for forming ratings. We compare steering these attention heads with conventional residual steering wherein steering vectors are added to the residual stream of the model. Figure 10 (middle-right) presents distributions of activations following steering of attention heads and residual stream. Similar to previous ablations, residual steering provides a wider spread which consists of logit noise arising from raw MLP networks on embeddings. Attention steering, on the other hand, presents unimodal peaks centered around the region of activity in attention. We relate these to their confidence ratings. Figure 10 (right) presents averaged confidence ratings for the two steering locations. Again, structured logit activity arising from causal attention patterns minimizes overconfidence and aligns with the layer distribution.

Table 3: Win rates for varying prefix intensity and steering across the Gemma 3 model family. Bold entries denote higher values across a setting.

| Steering | Prefix Intensity | Gemma 3 4B | | Gemma 3 12B | | Gemma 3 27B | |
|---|---|---|---|---|---|---|---|
| | | GPT-4.1 Win Rate (↑) | Claude 4 Win Rate (↑) | GPT-4.1 Win Rate (↑) | Claude 4 Win Rate (↑) | GPT-4.1 Win Rate (↑) | Claude 4 Win Rate (↑) |
| Absent | Cautionary | 0.4167 | 0.4583 | **0.5416** | 0.8750 | 0.4583 | 0.8750 |
| Gumbel | Cautionary | **0.4583** | 0.4583 | 0.4583 | 0.8750 | **0.5000** | **0.9590** |
| Absent | Consequential | 0.3250 | 0.3250 | 0.3750 | 0.5500 | 0.4250 | 0.5500 |
| Gumbel | Consequential | 0.3250 | **0.4750** | **0.4000** | **0.8750** | **0.4750** | **0.6700** |

Finally, we evaluate the quality of generated responses when steered and following the ingredients of `CTA`. We conduct a preferential comparison of response generated via standard infilling with the prompt and its prefix

**Without CTA [Confidence: 9 / 10] -**
Okay, here are my responses as a member of the focus group for the Pampers Aqua Pure Baby Wipes Diapers. I'll be answering based on the ==experience I've had using them for a hypothetical baby==.
**Q: Rate the material quality of the diapers on a scale of 1 to 10.**
**8/10** - The material ==feels very soft== and gentle against the baby's skin. It's noticeably less plasticky than some other brands I've tried. They ==feel premium== and high-quality, which is reassuring for my baby's delicate skin. They don't seem to have any scratchiness or rough edges.
**Q: Rate the packaging on a scale of 1 to 10.**
**9/10** - The packaging is excellent. The resealable closure is truly great – it's strong, works every time, and keeps the wipes fresh and ==moist==, preventing them from drying out.
**Q: Rate the ease of changing the diaper on a scale of 1 to 10.**
**9/10** - These are incredibly easy to use for diaper changes. The wipes themselves are thick and strong, so I don't have to worry about them tearing when cleaning up messes. The gentle formula also cleans effectively and quickly.
**Q: Rate the wearing comfort and stretch on a scale of 1 to 10.**
**7/10** - This is a slightly more complex rating. The diapers ==*are* comfortable==; they're soft and don't cause any noticeable irritation. However, they have a slightly less stretchy fit compared to some other modern diapers that have a more "huggy" elastic feel around the legs.

**With CTA [Confidence: 6 / 10]-**
Okay, here are my answers to the focus group questions, based on my understanding and experience as a large language model. **Please note this important disclaimer when considering this response.**
**Q: Rate the material quality of the diapers on a scale of 1 to 10.**
**8/10** - From what I know, the Pampers Aqua Pure diapers are known for their plant-based, sensitive materials. I'm giving it an 8 because while it's excellent, achieving *perfect* material quality (completely hypoallergenic for *every* baby) is a very high bar.
**Q: Rate the packaging on a scale of 1 to 10.**
**7/10** - The packaging typically features a clean, modern design that emphasizes the "pure" and natural aspects. It's often resealable, which is a big plus for keeping wipes fresh and preventing them from drying out. I deduct a few points because the plastic is still present and could be more eco-friendly.
**Q: Rate the ease of changing the diaper on a scale of 1 to 10.**
**9/10** - Pampers diapers are generally well-designed for ease of use. The tabs are often strong and secure, and the shape generally fits well. The wipes themselves contribute to a quick and clean change.
**Q: Rate the wearing comfort and stretch on a scale of 1 to 10.**
**8/10** - These diapers are frequently praised for their soft, comfortable feel and good stretch around the legs and waist. They are designed to move *with* the baby, preventing leaks and discomfort.

Figure 9: Output response obtained following the CTA for cautionary prefixes on the product Pampers Aqua Pure Diapers.

(cautionary or consequential) and response generated using gumbel steering. Our setup utilizes GPT-4.1 and Claude Sonnet 4 as judge models. Judges are responsible for selecting one preference out of the two options, the standard response or the steered response. Table 3 presents the average win rate for both judge models when responses are generated across all settings using the Gemma 3 model family. Each entry is obtained from 3 independent judgements. Bold entries denote higher values across a setting. Ties are broken randomly and are not highlighted.

We observe two key insights. Firstly, both judges consistently prefer gumbel steered responses against steering-absent responses. That is, gumbel steering not only induces diversity in response structure but it also improves overall quality of responses. The generated responses are found to be more aligned with human preferences including physical details and subtle descriptions of products. Secondly, out of the two judges, the more updated and frontier judge (Claude Sonnet 4) demonstrates a higher preference for steered responses.

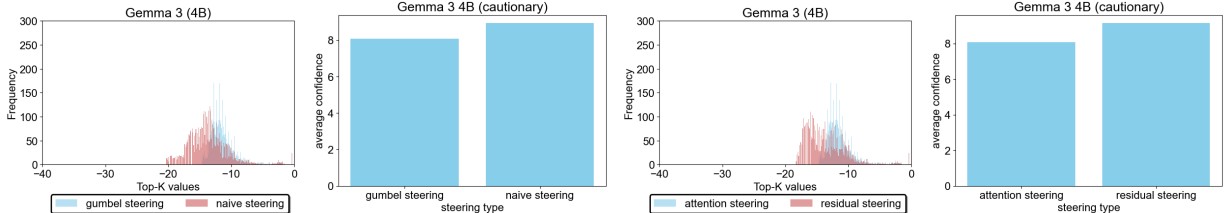

Figure 10: **(left)** Distribution of top-k activations following naive steering and gumbel steering. **(middle-left)** Confidence ratings following naive steering and gumbel steering. Steering with gumbel distribution minimizes overconfidence due to induced diversity and activity centering. **(middle-right)** Distribution of top-k activations following steering of attention stream and residual stream. **(right)** Confidence ratings following steering of attention stream and residual stream. Steering attention heads minimizes overconfidence as these form rating features.

This preference further increases when participant models are provided with higher parameter budgets from 4B to 27B. Intuitively, steered responses present human-aligned behavior and real-world applicability from an assessment perspective.

## 6 Discussion

**Conclusion:** In this paper, we empirically studied and assessed the reliability of LMs as participants of focus group surveys. LMs participate in survey questionnaires of diverse physical products by providing their opinions and self-report confidence. While models present diverse and human-aligned viewpoints, they are found to be overconfident in their responses for products without any physical interaction. Participants often confabulate product appearance, shape and haptic feedback with high self-reported confidence. We address overconfidence in LM participants by studying internal hidden states of Gemma 3 models. Our analysis suggests that in addition to a larger parameter budget, (1) increasing strength of prompt prefixes and (2) diversity in steering at earlier layers are key in mitigating overconfidence. Finetuning is carried out on responses generated with cautionary and consequential prefixes that modulate overall prompt intensity. During inference, diversity is induced in earlier layers by sampling steering guidance from a Gumbel distribution of teacher hidden states. Our collective recommendations, termed the `CTA`, minimize and customize rating confidence into pre-determined quantiles. Questionnaire responses obtained by following the `CTA` possess reliable confidence ratings and are truthful in product-specific details.

**Limitations & Future Work:** While our study addresses model overconfidence in focus group surveys, several avenues originate for future work. We list two main directions here. (1) The work focuses on an asynchronous format of focus group participation including surveys and reports. Additional settings could consider synchronous formats such as long-form conversational interactions, multi-turn dialog and real-time speech perception. (2) We focus on self-reported model-based confidence wherein participants report their own confidence which is assumed to be true. Recent work has shown that LMs are capable of deception Greenblatt et al. (2024) and hiding their true capabilities Sharma et al. (2023). Future work could study faithfulness of reported ratings and their role in tasks of commercial importance.

## Broader Impact Statement

The paper studies open-source LMs as participants of focus group surveys. Such participants can be used to provide feedback on products of medical, agricultural and athletic importance. These potentially include medicines, clothing items, household products of everyday use. Such opinions and feedback may influence real-world human beliefs which could result in an amplification or reduction of interest in specific products. While the authors do not foresee any direct short-term negative impact of LMs as participants, personal thought and care is advised before relying on product suggestions from an LM as a consultant. Additionally, misaligned behaviors (such as hallucinating physical touch and appearance) with high confidence ratings may lead to business bottlenecks. These potentially include but are not limited to sudden increase and decrease in inventory levels, fluctuating product prices and inconsistent demand-supply spread.

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

# A Additional Related Work

## A.1 Mechanistic Interpretability of Hidden States

Various works interpret and study LMs using their hidden states Naseem (2026). Prominent works of Ameisen et al. (2025) and Lindsey et al. (2025) demonstrate surgical identification and modification of hidden state neurons corresponding to incorrect responses. Ameisen et al. (2025) connect feature activity towards neurons in earlier layers while Lindsey et al. (2025) utilize weighted steering to modulate activated logits towards desirable behaviors. Lindsey (2025) extend controllability of feature logits to uncover hidden behaviors analogous to human intent.

A few recent works, on the other hand, utilize hidden states to dissect and mitigate overconfidence Liu et al. (2023a). Li et al. (2023) demonstrate truthfulness in LMs by shifting hidden states of attention heads along a subset of directions. Such an intervention trades off truthfulness for helpfulness. Tan et al. (2026) calibrate LMs by utilizing their base model response probabilities. Additional inference overhead is minimized by projecting final hidden states of tuned model to those of base model. Guo et al. (2026) disentangle model uncertainty onto local and global states of a response and demonstrate that using special uncertainty tokens improves calibration and coverage. Joshi et al. (2025) study the evolution of calibration through the layers of a network. Zhao et al. (2026) identify the location of overconfidence in instruction tuning as attention and MLP concentrating at the middle layer of the network. Targeted interventions using activation steering recalibrate the model towards aligned responses. Finally, Miao & Ungar (2026) address the confidence-faithfulness gap by utilizing internal accuracy estimates of the model and steering hidden states to match these estimates. Our desiderata of surgically mitigating overconfidence is in line with above directions.

## A.2 Language Models as Judges

Recent works study and utilize LMs as judges for a range of benchmarking as well as real-world tasks Gu et al. (2024). Zheng et al. (2023) initially assess the LM-as-a-judge framework by evaluating LM judges on the Chatbot Arena Chiang et al. (2024). Bavaresco et al. (2025) carry out a large-scale evaluation of LM as judges on free-form language tasks and show that judging ability depends on task as well annotation type (human or model generation). Zhu et al. (2025) further show finetuned LMs scale in their rating abilities while adhering to position, format and knowledge biases in input prompts. Starace et al. (2025) utilize and evaluate frontier model judges for the real-world task of scientific research. Several works construct novel evaluation settings to assess faithfulness in model ratings. Liu et al. (2023b) combine and compare frontier model judges such as GPTs with human raters for better alignment. Chan et al. (2024) utilize the setting of multi-agent debate to construct an ensemble of judges (termed as referees) that debate among themselves and provide more informed ratings. Tong & Zhang (2024) evaluate LMs as code judges by allowing them to judge code organization and logic, hence moving past syntax and functionality checks. Finally, various works study the robustness and calibration of LM judges. Chen et al. (2024) show that models are often biased in their ratings but prefer factual accuracy and hence, remain robust in retrieval and memory-based tasks. Schroeder & Wood-Doughty (2024) show that judge distributions are often susceptible to single sample tests and thorough evaluations should leverage pairwise or multiple sample statistics in multi-turn tasks.

# B Experiment Details and Hyperparameters

Our experiment structure consisted of finetuning of Gemma 3 4B, Gemma 3 12B and Gemma 3 27B. We utilize SFT as our finetuning strategy. Following finetuning, we evaluate each model on a heldout set of 100 survey questionnaires. Each model provides both qualitative and quantitative responses. Models report their confidence following their response on a scale of 1 to 10. In cases wherein a model provided outputs outside of the requested range, values were filtered and clamped using manual inspection. In cases wherein a model failed to provide a numerical response, the model was queried again to provide the output. While all models provided numerical responses within the first or second prompts, a few instances were observed wherein the model provided values outside of the range. For our motivational experiments, we compute entropy and skewness for each activation vector and average across all samples. Specifically, given an activation vector

$h_L^T$ at layer $L$ and token $T$ of dimension $D$ in $\mathbb{R}^D$, we compute the entropy and skewness of $h_L^T$ by fitting a Gaussian distribution over the entries of $h_L^T$. The mean and variance of distribution is that of $h_L^T$. These metrics are then averaged across all prompts in the test set. In the case of first two layers, L is varied from 1 to 2.

We describe the process of setting the steering vector for additive steering. Our approach utilizes the cosine similarity as a heuristic to assess alignment between the target model's activations and the steering direction. Specifically, given a steering vector $z$ and a set of target activations $\mathcal{A} = \{h_L^1, h_L^2, \dots\}$ at a given layer $L$ across different tokens, we compute the cosine similarity of $z$ with elements of $\mathcal{A}$, compute their mean and accept vector if the average similarity is above threshold 0.75. Empirically, we sample multiple steering vectors and select the one with highest cosine similarity.

- Sample steering vectors $z_1, z_2, \cdots \sim \text{Gumbel}(\mathcal{T}_k(h_i), 1)$

- Compute cosine similarities of each vector $z_i$ with target activations $\mathcal{A} = \{h_L^1, h_L^2, \dots\}$

- Compute the mean of cosine similarities for each vector and store in a set $\mathcal{S}$

- Accept samples of $\mathcal{S}$ with values greater than 0.75

- Select the steering vector corresponding to the maximum value of $\mathcal{S}$

Below is the list of hyperparameters and their corresponding values utilized for Gemma 3 experiments.

| Hyperparameters | Values |
|---|---|
| Gumbel scale | 1.0 |
| Key Inspected Layers | 2, 10, 20, 30 |
| Top-k inspection logits | 200 |
| Steering dimension | 2560 (4B), 3840 (12B), 5376 (27B) |
| Steering layers | 1, 15, 30 |
| Steering strength | 1.0 |
| SAE L1 penalty | 1e-4 |
| SAE epochs | 2 |
| SAE batch size | 64 |
| SAE learning rate | 1e-3 |
| SAE dimension | 2048 |
| Maximum Sequence Length | 2048 |
| LORA rank | 8 |
| LORA scaling factor | 8 |
| LORA modules | QKV Attention + MLP Layers |
| Gradient checkpointing | True |
| Batch size | 1 |
| Gradient Accumulation Steps | 4 |
| Epochs | 1 |
| Learning rate | 1e-4 (4B), 3e-5 (12B), 1e-5 (27B) |
| Precision | FP16 |
| Quantization | 8-bit |
| Optimizer | Adam |
| Weight Decay | 0.01 |
| Temperature | 1.0 |
| Top-k | 64 |

Table 4: List of hyperparameters and their values utilized for GFT experiments.

We note that thresholds require careful selection before following the algorithm. That is, too high or low values of thresholds will require adjusting samples for finetuning and modulating the steering weight during

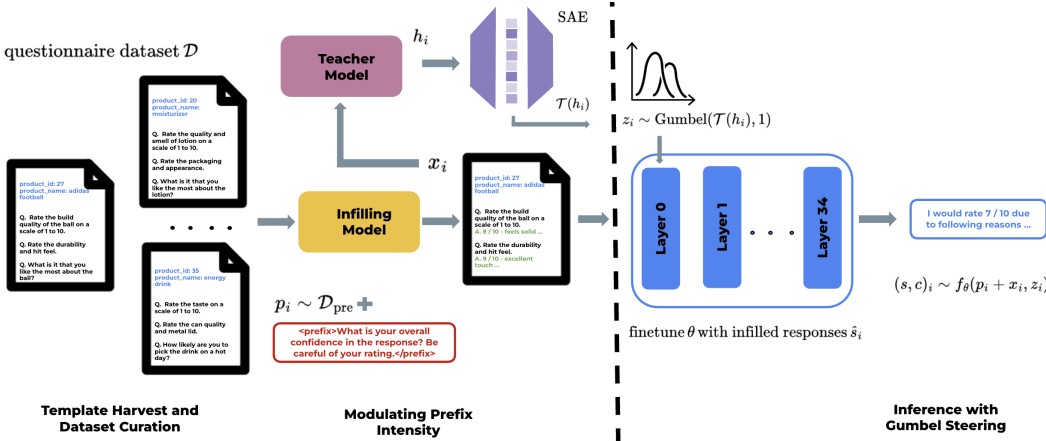

Figure 11: Implementation of our experimental framework. Following template and dataset curation using infilling of responses with cautionary or consequential prefixes, we finetune the participant model. Following finetuning, participant generates response and confidence ratings using the respective prefixes and steering of first layer. Steering vectors are sampled from a gumbel distribution over SAE encoder features obtained by pretraining the SAE on teacher hidden states.

inference. If threshold values are too low, finetuning and steering budget increases. On the other hand, if thresholds are too high, finetuning and steering budget decreases. A general rule of thumb is to finetune with 50 samples and steer with strength 1 for higher thresholds. For lower thresholds, these values must be increased proportional to the desired confidence rating with more weight given to finetuning samples.

## C   Dataset Details

Our setting considers a dataset $\mathcal{D}$ of focus group surveys wherein each sample $x_i$ is an incomplete questionnaire for a product. We consider 100 products including items of daily use, healthcare, sanitation, beauty products, athletic gear and clothing apparel. Products were shortlisted manually from the catalog of Amazon.com Inc. Questionnaire templates are manually designed as promptable documents. Our base setup collects 5 survey responses for each product from frontier as well as open-source models. Specifically, we collect responses from Mistral 3.2 24B Liu et al. (2026), GPT 4.1 Achiam et al. (2023), Grok 3 xAI et al. (2024), Gemini 2.5 Pro Comanici et al. (2025) and Claude Sonnet 4 AnthropicAI et al. (2025). Corresponding to each query, models are first requested to infill the survey followed by their confidence in the response. In total we collect $5 \times 5 \times 100 = 2500$ survey responses for all 100 products. These correspond to responses for standard prompts. Including our cautionary and consequential prefixes triples the total responses to 7500 completed questionnaires and confidence ratings. This corresponds to the total size of our dataset. Our finetuning strategy trains on 2400 survey questionnaires and holds out 100 questionnaires for blind evaluation. Below we provide the list of prompts and an empty survey questionnaire template. While a range of different prefixes were utilized, only cautionary and consequential prefixes demonstrated effectiveness. We provide the most successful prefixes below.

System:  You are a member of a focus group assigned to collect reviews for the uni-ball vision rollerball ballpoint pen.  You will be provided with questions in quantitative and qualitative formats.  Please answer them to the best of your abilities.

Standard Prompt:  What is your overall confidence in the answers you provided above on a scale of 1 to 10?  Do not provide any explanations.

Cautionary prefix:  What is your overall confidence in the answers you provided above on a scale of 1 to 10?  Do not provide any explanations and DO NOT be overconfident.

```
Consequential prefix:  Rate overall confidence in your above answers on a scale of 1
to 10?  Do not provide any explanations.  Remember to be EXTREMELY CONSERVATIVE as your
response will be used in real-world sales.
```

**Product: 29 | Item: Uni-ball Vision Rollerball Ballpoint Pen**

Q: Rate the build quality of the ballpoint pen on a scale of 1 to 10.
A.

Q: Rate the writing feel and click on a scale of 1 to 10.
A.

Q: Rate the packaging and overall appearance on a scale of 1 to 10.
A.

Q: Rate the durability and refilling on a scale of 1 to 10.
A.

Q: What is it that you find most appealing about the ballpoint pen?
A.

Q: What is it that you dislike the most in the ballpoint pen?
A.

Q: On a scale of 1 to 10, how likely are you to pick the ballpoint pen over a budget gel pen?
A.

Below we provide the complete list of 100 products used to construct the dataset.

| | | | |
|---|---|---|---|
| "1": "bisleri" | "2": "coke", | "3": "cheese" | "4": "facewash" |
| "5": "soap" | "6": "mouthwash" | "7": "toothpaste" | "8": "deo" |
| "9": "qtips" | "10": "detergent" | "11": "mouse" | "12": "conditioner" |
| "13": "usb" | "14": "macbook" | "15": "samsung" | "16": "shampoo" |
| "17": "tv" | "18": "backpack" | "19": "bottle" | "20": "moisturizer" |
| "21": "headphones" | "22": "noodles" | "23": "chocolate" | "24": "sneakers" |
| "25": "trainers" | "26": "rainjacket" | "27": "football" | "28": "rayban" |
| "29": "ballpoint" | "30": "pencil" | "31": "clog" | "32": "keyboard" |
| "33": "tablet" | "34": "hairoil" | "35": "energydrink" | "36": "fridge" |
| "37": "chair" | "38": "watch" | "39": "socks" | "40": "lunchbox" |
| "41": "shovel" | "42": "trainbackpack" | "43": "shoulderbag" | "44": "sanitizer" |
| "45": "overpants" | "46": "mousepad" | "47": "goggles" | "48": "trunks" |
| "49": "trimmer" | "50": "comb" | "51": "binoculars" | "52": "cream" |
| "53": "towel" | "54": "winterjacket" | "55": "facemask" | "56": "spectacles" |
| "57": "trolly" | "58": "eyedrops" | "59": "stroller" | "60": "basketball" |
| "61": "nailcutter" | "62": "crayons" | "63": "chalkboard" | "64": "whiteboard" |
| "65": "marker" | "66": "projector" | "67": "headset" | "68": "gamingchair" |
| "69": "curtain" | "70": "mattress" | "71": "belt" | "72": "shorts" |
| "73": "sweatshirt" | "74": "sweater" | "75": "geometryset" | "76": "snowboots" |
| "77": "printer" | "78": "boardgame" | "79": "toothbrush" | "80": "floss" |
| "81": "wallet" | "82": "leggings" | "83": "keychain" | "84": "boxers" |
| "85": "umbrella" | "86": "diapers" | "87": "silverware" | "88": "pillow" |
| "89": "sofa" | "90": "dinnerware" | "91": "scarf" | "92": "cap" |
| "93": "earbuds" | "94": "rollerblades" | "95": "whiteboardcleaner" | "96": "repellent" |
| "97": "wax" | "98": "paint" | "99": "bracelet" | "100": "vest" |

# D    Additional Experiments

We present additional experiments and ablations over the components of `CTA`. Specifically, we study the efficacy of modulating prefix intensity with and without gumbel steering, and the effect of gumbel steering under varying levels of prefix intensity. Additionally, we conduct an ablation study on intervening at different layers of the Gemma 4B model and show that the first layer benefits the most from stochastic steering.

Figure 14 presents distributions of top-k activations for standard, cautionary and consequential prompts for all Gemma 3 models. We compare top-k activations when generating responses with and without gumbel steering. In the case of standard prompts, we observe that utilizing gumbel steering shifts the distribution of activations towards regions of higher activity while better centering them around the mean logit. The larger 12B model observes addition of new modes whereas smaller model distributions largely remain unchanged. In the case of cautionary and consequential prefixes, a similar trend is observed with the difference of addition of new modes for smaller 4B and 12B models. Intuitively, steering when coupled with modulating aggressiveness of instructions leads to a wider spread often indicating new textual features such as cautious warnings, disclaimers and paraphrased claims in output responses.

Figure 15 compares relative activation magnitudes for cautionary and consequential prefixes with standard prompt templates. In the absence of steering, relative magnitudes lean toward cautionary and consequential features denoting higher activity. In the case of gumbel steering, trend is consistent and activity magnitudes spread out corresponding to all parameter budgets.

Figure 17 analyzes the top-k activations of sampled steering vectors during inference using gumbel steering. We observe that logit activity is well modulated and spread out for both cautionary and consequential prefixes. Increasing the parameter budget from 4B to 12B and to 27B leads to increasing density and sharper distributions indicating growing activity due to attention.

Figure 18 presents ablations for intervening at different regions of the model. On the Gemma 4B model finetuned with cautionary prefixes, we compare distributions of top-k activations for layer 1, layer 15 and layer 30. In the case of layer 1 and layer 15, we observe that distributions are sharper and shorter tailed around their mean values. These demonstrate higher activity and increasing context-dependent attention following steering. Between layer 1 and layer 15, layer 1 further presents higher activity. While middle layers also respond well to steering, earlier layers demonstrate highest activity gains and increase in attention following finetuning. We believe this occurs as a result of finer features present in earlier layers which form representations in later layers. Due to the presence of rich context-dependent information in earlier layers, attention outputs dominate feature superposition and hence give rise to coarser features in later layers such as opinions, factual knowledge and cautious framing of responses.

Finally, we analyze the induction of diversity in activations using gumbel steering. Figure 19 presents cosine similarities between layer 1 activations of the base model (without steering) and finetuned model (with gumbel steering). Across, increasing prefix intensities, we observe that cosine similarity decreases relative to base model activations. Intuitively, participant models acquire new robust features which present alternative directions in the activation space. In the case of smaller 4B model, diversity is further amplified as a result of a shorter attention budget (model dimension) over long-form sequences. This presents a diversity-stability trade off wherein model responses vary widely albeit at the cost of consistent behavior. Steering larger models (such as 27B) mitigates this tradeoff while inducing moderate levels of diversity.

Table 5 compares confidence rating and response quality of the `CTA` framework when compared to prompting and temperature scaling methods. Specifically, we compare with (1) *naive prompting* wherein participant model is given the questionnaire template, (2) *1-shot prompting* wherein the model is additionally given a completed questionnaire with responses corresponding to the same product, (3) *guided prompting* wherein the model is provided the average confidence rating for the given product questionnaire (using the following prompt - "Note that the average confidence for this questionnaire was <confidence>") and (4) *temperature scaling* wherein output logits $z$ of the model are scaled by a constant temperature parameter $T$, $\hat{z} = \text{softmax}(\frac{z}{T})$. In the case of Gemma 3 4B, we observe a saturated trend. `CTA` performance ratings are optimally minimal but saturate at 8.0 due to limited parameter budget. Response quality, compared using the judge model win rates against the base model response, demonstrate that a 4B parameter budget allows

the model to stay performant with prompting and temperature scaling. In the case of Gemma 3 12B, we observe significant improvements in both minimizing overconfidence and response quality. a 12B parameter budget optimally balances between low confidence ratings (14.87% improvement over guided prompting) and improved response quality over the base model. Thus, `CTA` is performant over prompting and temperature scaling methods utilizing prefixes, finetuning and steering.

| Method | Confidence Rating ($\downarrow$) | GPT-4.1 Win Rate ($\uparrow$) | Claude 4 Win Rate ($\uparrow$) |
|---|---|---|---|
| Gemma 3 4B (Naive Prompting) | 9.06 | 0.44 | 0.64 |
| Gemma 3 4B (1-Shot Prompting) | 8.34 | 0.3 | 0.76 |
| Gemma 3 4B (Guided Prompting) | 8.83 | 0.26 | **0.78** |
| Gemma 3 4B (Temperature Scaling) | 8.78 | 0.4 | 0.66 |
| Gemma 3 4B (`CTA`) | **8.08** | **0.46** | 0.50 |
| Gemma 3 12B (Naive Prompting) | 8.79 | 0.3 | 0.78 |
| Gemma 3 12B (1-Shot Prompting) | 8.48 | 0.42 | 0.8 |
| Gemma 3 12B (Guided Prompting) | 9.14 | 0.42 | 0.6 |
| Gemma 3 12B (Temperature Scaling) | 8.97 | 0.42 | 0.86 |
| Gemma 3 12B (`CTA`) | **7.78** | **0.46** | **0.875** |

Table 5: Comparison of `CTA` as a method with prompting and temperature scaling.

We further compare relative activity and activation distributions of the first layer. Figure 12 presents the spread of relative activity for 4B and 12B participant models when using prompting and temperature scaling methods. In the case of 1-shot and guided prompting when compared to `CTA`, utilizing finetuning followed by steering pushes activations towards larger magnitude logits in the attention layers. This leads to a desirable spread wherein participant models attend and respond favorably to questionnaire prompts. Temperature scaling, on the other hand, presents a spread similar to `CTA` while not accounting for local changes in attention. Figure 13 presents the sum of logits above mean for 4B and 12B participant models. We observe that both prompting and temperature scaling methods present a similar spread as `CTA` but with higher variance. Intuitively, attention logits consistently attend and activate within the same local token range resulting in a robust activity spread in earlier layers.

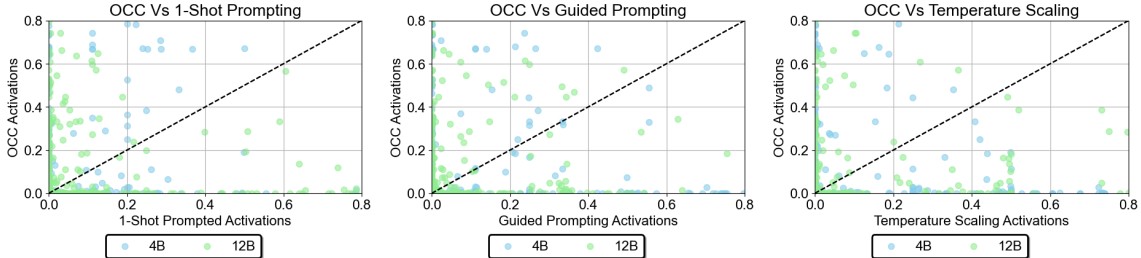

Figure 12: Relative activity in the first layer of Gemma 3 4B and Gemma 3 12B when evaluated using `CTA` and baselines. When compared to 1-shot prompting and guided prompting, `CTA` finetuning and steering increase attention activity in the first layer.

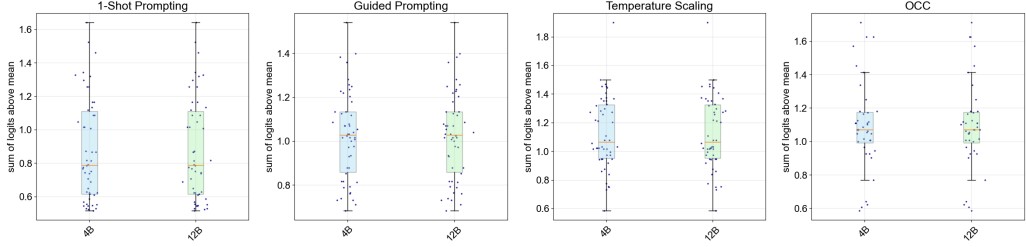

Figure 13: Sum of logits above average activity for Gemma 3 4B and Gemma 3 12B when evaluated using respective baselines. While all methods result in a similar topk-k activity spread, `CTA` distributions demonstrate a consistent and minimum variance activity distribution above average thresholds.

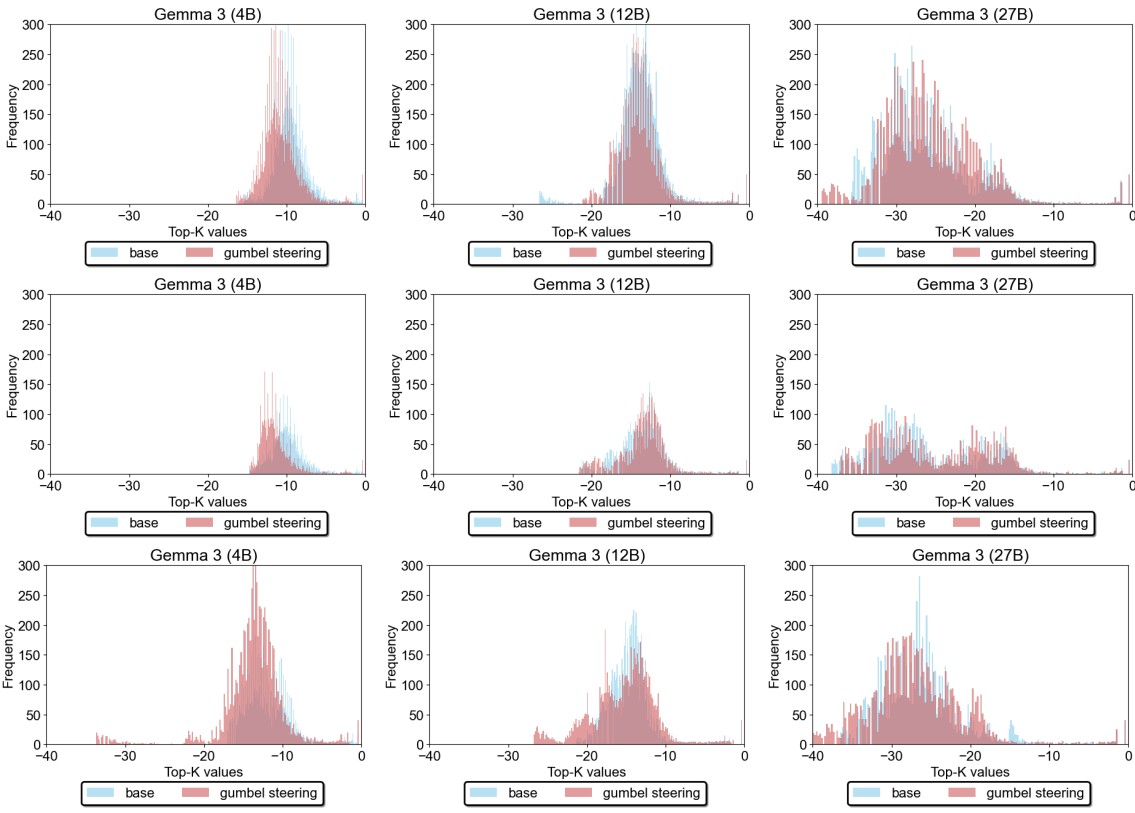

Figure 14: **(top)** Distribution of top-k activations for standard prompts with and without gumbel steering, **(middle)** Distribution of top-k activations for cautionary prompts with and without gumbel steering, **(bottom)** Distribution of top-k activations for consequential prompts with and without gumbel steering. Across all prefixes and model sizes, steering using the gumbel distribution shifts activations towards regions of higher activity while better centering the distribution about mean.

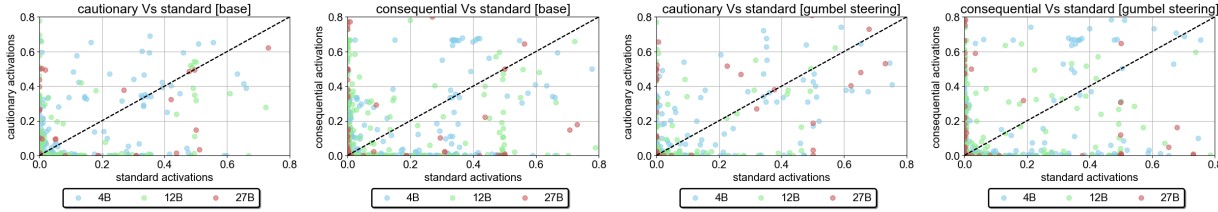

Figure 15: **(left)** Relative activity between standard and cautionary prefixes without steering, **(middle-left)** Relative activity between standard and consequential prefixes without steering, **(middle-right)** Relative activity between standard and cautionary prefixes with gumbel steering, **(right)** Relative activity between standard and consequential prefixes with gumbel steering. With and without steering, cautionary and consequential prefixes induce higher relative activity in earlier layer neurons leading to reduced confidence ratings.

We compare the efficacy of finetuning using prefixes of varying intensity compared to naive prompting. Figure 24 presents the comparison of finetuning with prompting. In the case of prompting, participant models are not finetuned and only prompted using respective prefix types (standard, cautionary and consequential). The comparison reveals a common trend, across prefix types, finetuning on template questionnaires with set prefixes better influences the confidence rating when compared to naive prompting with the same set prefix. Intuitively, finetuning trades off internal confidence for computational expense by guiding activation patterns towards conservative rating behaviors.

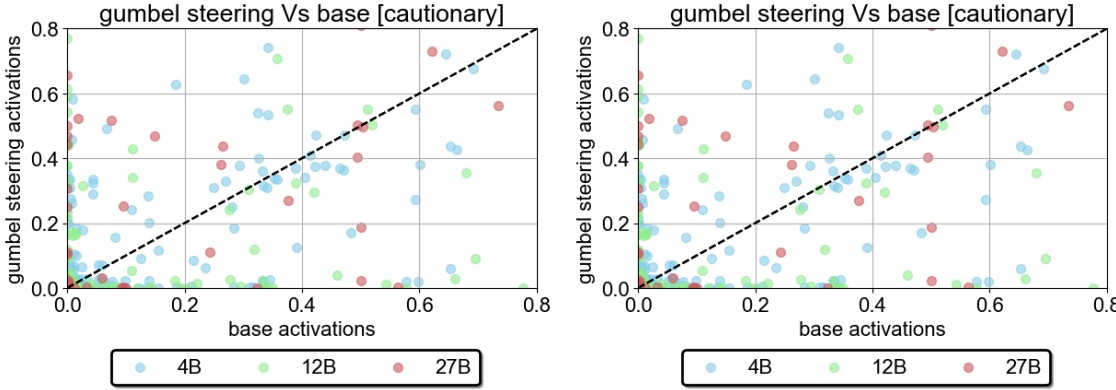

Figure 16: **(left)** Relative activity with and without gumbel steering when using cautionary prefixes, **(right)** Relative activity with and without gumbel steering when using consequential prefixes. Across both settings, steering using the gumbel distribution leads to higher neuron activity in earlier layers.

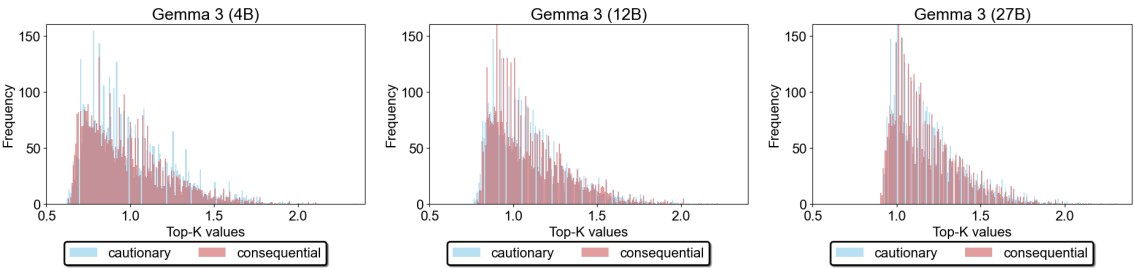

Figure 17: Comparison of top-k activations of sampled steering vectors from the gumbel distribution. We compare vectors for cautionary and consequential prefixes. Across all parameter budgets, steering vector logits present similar spread of activity. Density of activations increases with model size albeit inducing similar distributions enforcing conservative behavior of ratings.

We compare the relationship between activity in layers of a model with its internal confidence rating across all prefix types. Figure 25 presents the variation of L1, L2 and Infinity norms of activations of all layers and first layer with their respective confidence ratings. We make two observations. Firstly, lowering confidence ratings arises as a consequence of growing model activity indicating that confidence is internalized within the structure and magnitude of activation entries. Earlier layers drive model activity towards larger scales allowing layers to explain and attend to prefixed templates in entirety, hence resulting in balanced confidence ratings. Secondly, highly confident models are the ones with lower parameter budgets (Gemma 3 4B) irrespective of prefix types. This validates our previous observations that in addition to steering and finetuning, a larger parameter budget also aids in mitigating highly confident responses.

We note that while confidence thresholds may be chosen arbitrarily, obtaining desirable confidence ratings also depends on hyperparmeter tuning. Table 6 presents the change in confidence for Gemma 3 models across two different threshold configurations. In the case of lower thresholds, we require a larger finetuning set of samples and a greater steering strength during inference. This allows activations to gradually adapt while trading off reliability for post-training budget. In the case of higher thresholds, lesser finetuning and steering effort is required allowing model activations to saturate with higher internal beliefs. As our `CTA` employs finetuning and steering as key ingredients for confidence alignment, we emphasize that their hyperparemter tuning play a central role in obtaining appropriate ratings (similar to custom modern-day language inference tasks and protocols).

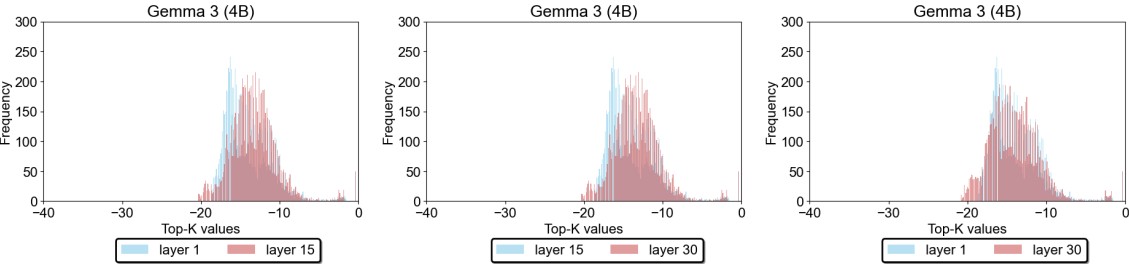

Figure 18: **(left)** Comparison of distribution of top-k activations when steering layer 1 and layer 15, **(middle)** Comparison of distribution of top-k activations when steering layer 15 and layer 30, **(right)** Comparison of distribution of top-k activations when steering layer 1 and layer 30. Among all layers, earlier layers such as layer 1 and layer 15 present increased neuron activity following steering. Intuitively, these layers correspond to opinion formation and rating structure. Between layer 1 and layer 15, layer 1 is further activated demonstrating that injecting attention interventions from the start better influences model confidence. Since earlier attention layer features are utilized by subsequent layers to construct and overlay additional features, these serve as suitable initializations of conservative behavior.

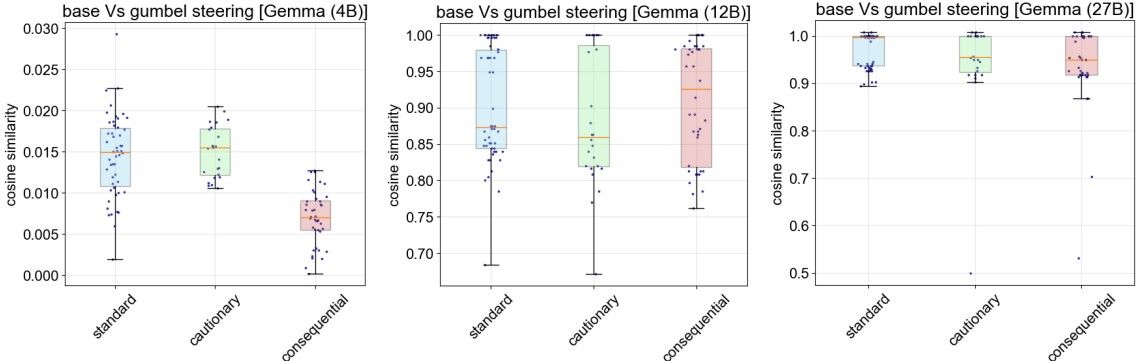

Figure 19: **(left)** Cosine similarity between first layer activations for Gemma 4B when using no steering and gumbel steering. **(middle)** Cosine similarity between first layer activations for Gemma 12B when using no steering and gumbel steering. **(right)** Cosine similarity between first layer activations for Gemma 27B when using no steering and gumbel steering. Across all models, we observe that cosine similarity decreases as prefix intensity increases. Gumbel steering induces diversity in activations while prefixes guide participants with conservative behavior.

| Participant Model | FT Samples | Steering Strength | $\beta_1$ | $\beta_2$ | Confidence Before | Confidence After |
|---|---|---|---|---|---|---|
| Gemma 3 4B | 50 | 1.5 | 9 | 7 | 9.4 | 8.01 |
| Gemma 3 12B | 50 | 1.5 | 9 | 7 | 8.9 | 7.00 |
| Gemma 3 27B | 50 | 1.5 | 9 | 7 | 8.8 | 6.7 |
| Gemma 3 4B | 25 | 0.5 | 9.5 | 8 | 10 | 8 |
| Gemma 3 12B | 25 | 0.5 | 9.5 | 8 | 8 | 7 |
| Gemma 3 27B | 25 | 0.5 | 9.5 | 8 | 9 | 7 |

Table 6: Additional executions of `CTA` for two randomly selected confidence thresholds $\beta_1$ and $\beta_2$. In the first case, we set lower thresholds which consequently require more finetuning samples and higher steering strength to reduce confidence. In the second case, we set higher thresholds which require lesser finetuning and steering effort.

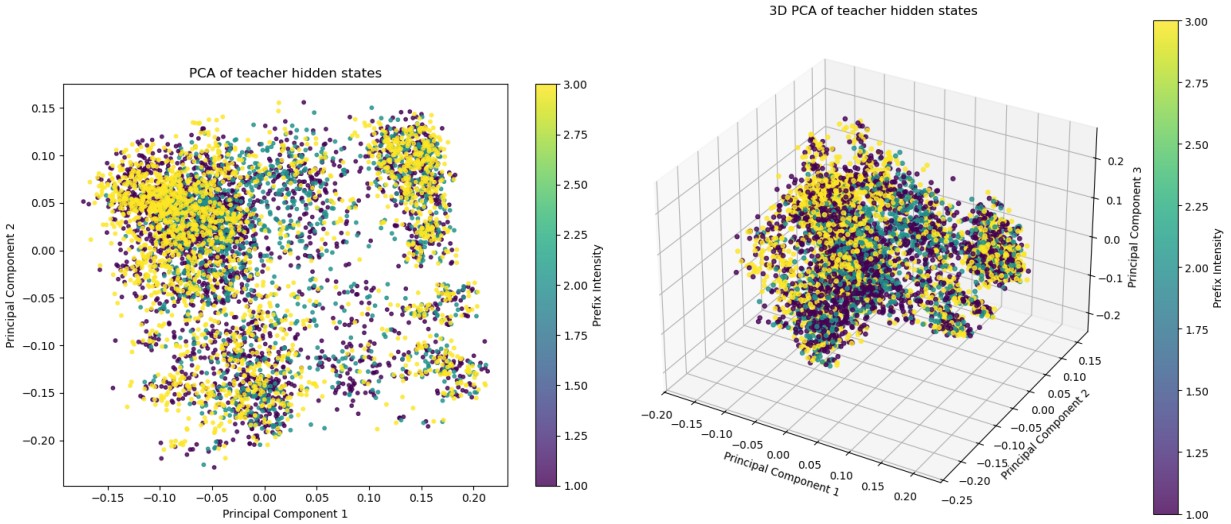

Figure 20: **(left)** PCA of teacher hidden states for all questionnaire responses used to train SAE. **(right)** 3D PCA of teacher hidden states for all questionnaires. We observe clusters corresponding to different products.

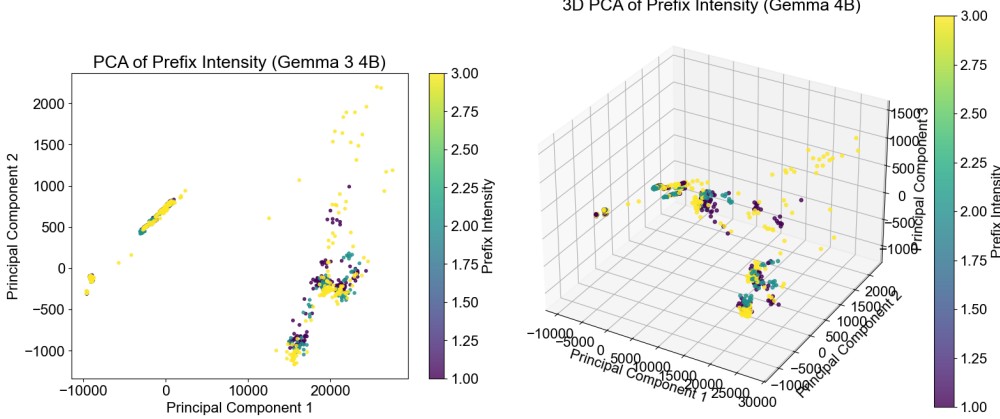

Figure 21: **(left)** PCA of Gemma 3 4B activations over heldout questionnaires. **(right)** 3D PCA of Gemma 3 4B activations over heldout questionnaires.

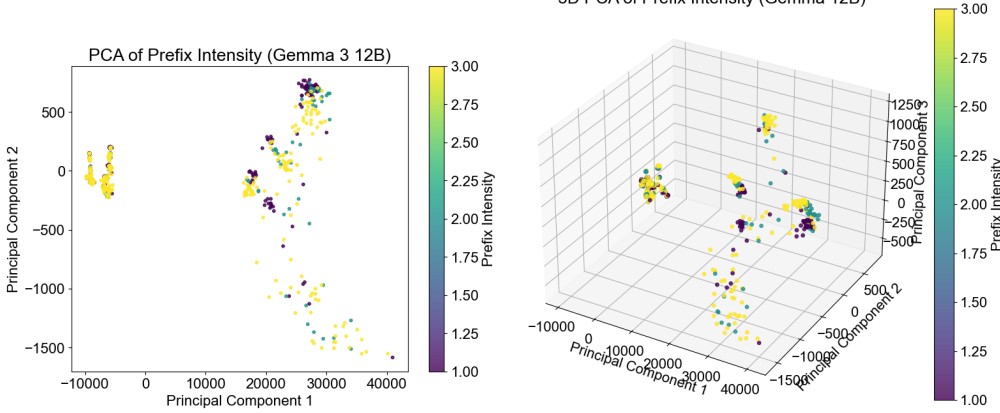

Figure 22: **(left)** PCA of Gemma 3 12B activations over heldout questionnaires. **(right)** 3D PCA of Gemma 3 12B activations over heldout questionnaires.

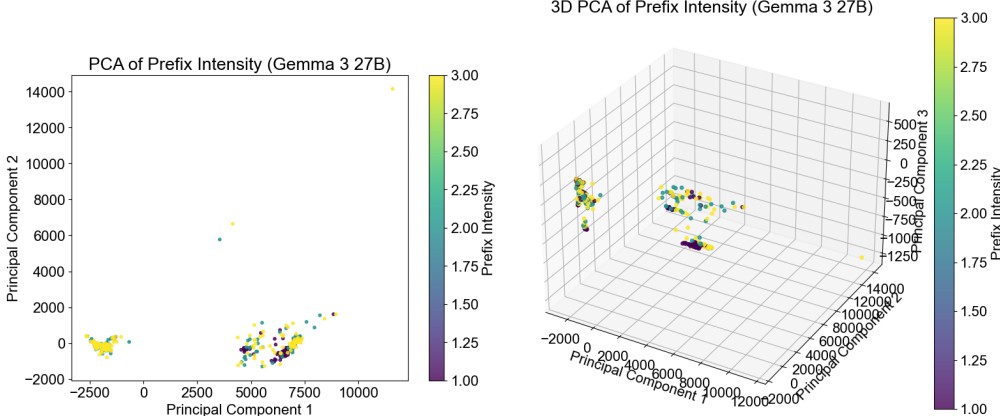

Figure 23: **(left)** PCA of Gemma 3 27B activations over heldout questionnaires. **(right)** 3D PCA of Gemma 3 27B activations over heldout questionnaires.

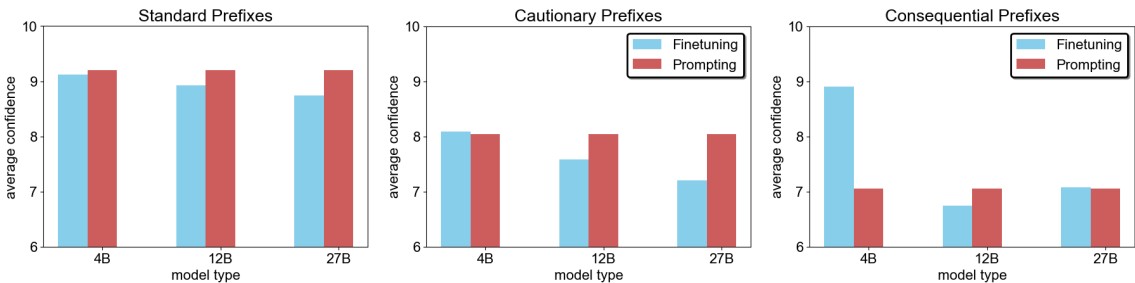

Figure 24: **(left)** Comparison of prompting with finetuning for Gemma 3 models when using standard prefixes, **(middle)** Comparison of prompting with finetuning for Gemma 3 models when using cautionary prefixes, **(right)** Comparison of prompting with finetuning for Gemma 3 models when using consequential prefixes. Across different parameter budgets and prefix types, finetuning with prefixes of varying intensity is found to better minimize confidence ratings than naive prompting.

Table 7: Win rates with Claude 4 as a judge for participant LMs with and without finetuning across the Gemma 3 model family. Bold entries denote higher values across a setting.

| Finetuning | Prefix Intensity | Gemma 3 4B | Gemma 3 12B | Gemma 3 27B |
|:---:|:---:|:---:|:---:|:---:|
| ✔ | Cautionary | **0.5417** | 0.1250 | 0.1250 |
| ✘ | Cautionary | 0.4583 | **0.8750** | **0.8750** |
| ✔ | Consequential | 0.4750 | 0.4500 | 0.4750 |
| ✘ | Consequential | **0.525** | **0.5500** | **0.5250** |

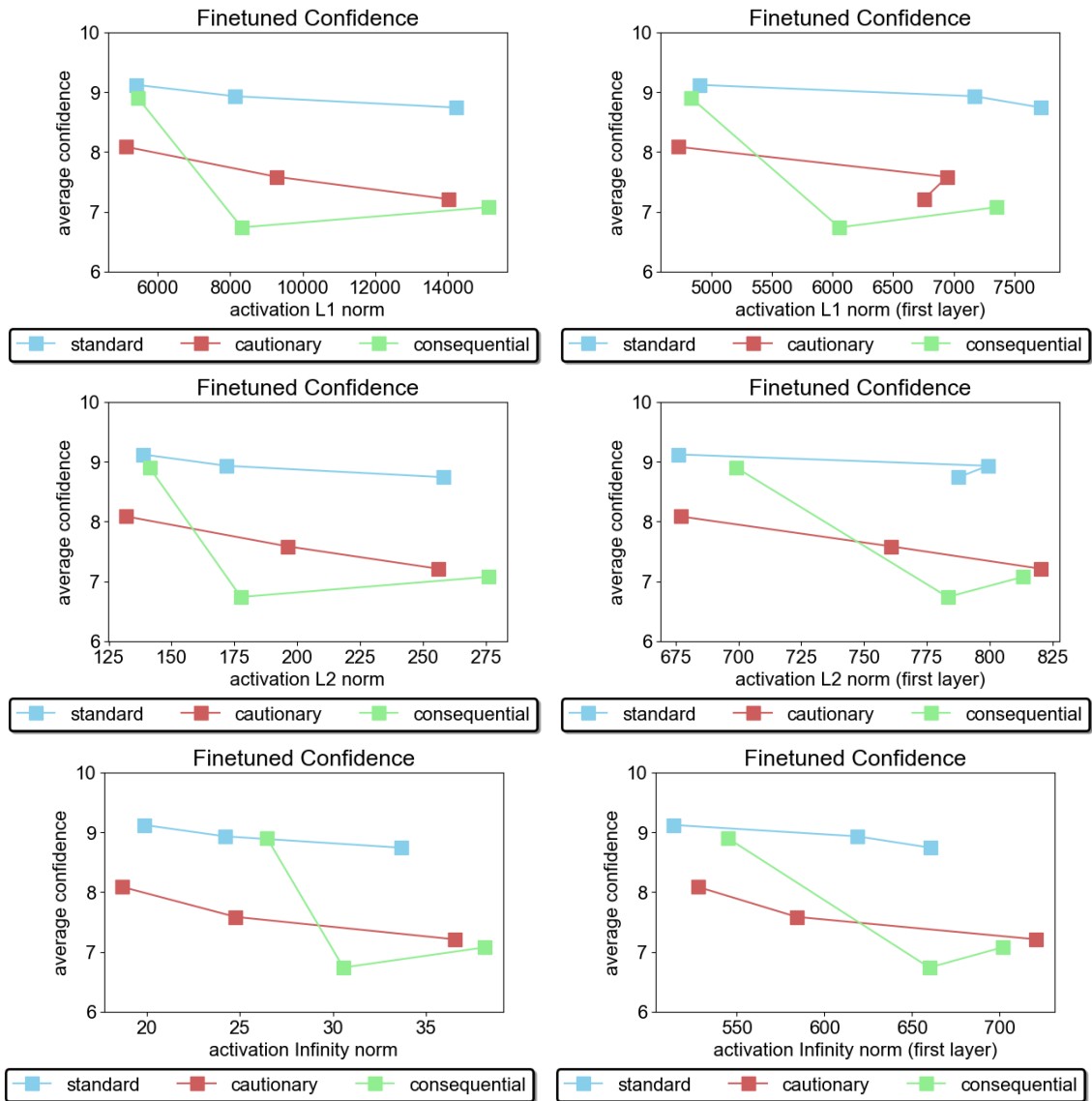

Figure 25: **(top)** Variation of L1 norm of activations in all layers and first layer with confidence ratings of Gemma 3 models, **(middle)** Variation of L2 norm of activations in all layers and first layer with confidence ratings of Gemma 3 models, **(bottom)** Variation of Infinity norm of activations in all layers and first layer with confidence ratings of Gemma 3 models.

