# OpenReview forum: "Understanding and Mitigating Overconfidence in Focus Group Surveys"
_TMLR — Under review for TMLR_

### Review · Reviewer_Pt7T · 2026-06-21

**Summary Of Contributions:**

This paper studies the overconfidence phenomenon of language models in simulated Focus Group Surveys. The authors demonstrate that the choice of prompt prefix and steering guidance at earlier layers are pivotal in mitigating overconfidence. They propose the Over-Confidence Checklist (OCC), a framework that aids in minimizing and customizing rating confidence into pre-determined quantiles. Empirical experiments show that the OCC leads to reliable confidence ratings while grounding responses in truthful, product-specific details rather than confabulations.

**Additional Comments:**

* On page 2, second line from the bottom: "ormpt intensity" should be "prompt intensity."

**Audience:**

Yes

**Audience Explanation:**

Focus Group Surveys are an emerging and important application for LMs, and model overconfidence is a critical topic in this domain. Both the experimental observations regarding LM internal states and the proposed Over-Confidence Checklist offer interesting insights for the research community.

**Broader Impact Concerns:**

As suggested by the authors, this paper studies focus group surveys generated by LMs. While they "do not foresee any direct short-term negative impact of LMs as participants, personal thought and care is advised before relying on product suggestions from an LM as a consultant," especially since such generated feedback can wrongfully influence real-world human beliefs about specific commercial products.

**Claims And Evidence:**

Yes

**Claims Explanation:**

The authors provide extensive experiments. They also plot their results clearly and illustrate their findings well throughout the paper.

**Requested Changes:**

I am not an expert in this particular area, but I have a few concerns:

* **Why is overconfidence a bad thing after all?** The confidence is just self-reported by the model. Is there a particular reason to penalize it so heavily? The paper notes that models confabulate product appearance and haptic feedback (e.g., the soft touch of a diaper)---things they cannot physically experience---with high confidence. It would strengthen the paper to elaborate more explicitly early on why this high-confidence hallucination/confabulation is dangerous for downstream commercial goals and human belief formation.

* **The Over-Confidence Checklist looks a bit too complicated.** The pipeline requires dynamic prefixing based on activation thresholds, model finetuning, training a Sparse Auto-Encoder (SAE) on a teacher model, and applying inference-time Gumbel steering. The observed improvements may be a result of over-engineering instead of a fundamental and accessible fix. Have the authors compared this heavily engineered pipeline to much simpler calibration techniques, such as standard RLHF/DPO calibration, basic temperature scaling, or simple few-shot system prompting?

---

> ### Author Response · Authors · 2026-07-16
> **Response To Reviewer Pt7T (1 / 2)**
>
> We thank the reviewer for providing feedback which is of utmost value to our work. Below we aim to address your concerns with changes in the revision marked in $\color{blue}{\text{blue}}$.
>
> > Why is overconfidence a bad thing after all? The confidence is just self-reported by the model. Is there a particular reason to penalize it so heavily?
>
> We recall that similar to humans, participant models respond with diverse and detailed opinions. Naturally, responses are aligned with human intent. However, contrary to humans, models do not interact with products or observe their physical traits directly in an extended setting. This leads to a fundamental issue wherein limited textual/visual information drives stronger beliefs about products. Participants are found to be overconfident in their responses for products without any physical interaction. Models often confabulate product appearance, shape and haptic feedback with high self-reported confidence. That is, models overly rely on distilled product-specific descriptions rather than forming their opinions through mediation or thinking.
>
> In the case of objective tasks such as ranking and QA, ground-truth confidence ratings may be acquired from larger accurate models or human judges. However, in subjective evaluation tasks self-reported confidence and model’s internal beliefs become the primary measures. While different models report ratings on a different scale while considering different product and usage-specific aspects, their overall ratings still tend to the higher side when compared to chance outcomes (5 / 10 rating as per Bernoulli trials) or honest refusals.
>
> We observe that high confidence ratings in itself do not indicate an adverse condition. When combined with (1) hallucinated responses (such as specific descriptions of touch, smell and motion), (2) a lack of refusal ability and (3) consistently high ratings excessively aligned with human intent, these lead to overconfidence in subjective tasks. Thus, we aim to tackle overconfidence not simply as a means of higher self-reported values. Rather, overconfidence is tackled when observed along with the above three components of hallucinations, limited refusals and excessively sycophantic alignment.
>
> > It would strengthen the paper to elaborate more explicitly early on why this high-confidence hallucination/confabulation is dangerous for downstream commercial goals and human belief formation.
>
> Thank you for the suggestion. Following the reviewer’s feedback, we have extended our discussion in Section 1 and the Broader Impact Statement. Specifically, we emphasize that misaligned behaviors such as haptic feedback and sense of touch and smell, supported by highly confident internal beliefs, pose commercial and marketing risks for organizations. These include but are not limited to inventory mismanagement, price slippage and noisy demand-supply patterns. High confidence ratings and overly relying on hallucinated responses may lead to business bottlenecks. These potentially include sudden increase and decrease in inventory levels, fluctuating product prices and inconsistent demand-supply spread.
>
> > The Over-Confidence Checklist looks a bit too complicated. The pipeline requires dynamic prefixing based on activation thresholds, model finetuning, training a Sparse Auto-Encoder (SAE) on a teacher model, and applying inference-time Gumbel steering. The observed improvements may be a result of over-engineering instead of a fundamental and accessible fix.
>
> Thank you for raising the concern. We recall that the checklist consists of only 3 standard components -
>
> * Prefixing - Questionnaire templates are prefixed with cautionary or consequential prompts in order to inform the model of the importance of its confidence ratings. Specifically, we do not train external models to elicit prefixes or prefix classifiers to adaptively select a subset. Prefixes are selected from one of the two categories based on confidence ratings and their respective threshold bins.
> * Finetuning - We finetune participant models only on infilled responses. These responses do not include base confidence ratings or any guidance relating to their evaluation protocols. Conventional SFT finetuning is carried out to distill product-agnostic knowledge and writing style which allows participant models to adopt a reliable response framing strategy.
> * Steering - Post-finetuning, gumbel steering is carried during evaluation. As before, steering vectors are not generated or sampled per-token. We only sample the steering vector once before evaluation, following its application to the earliest attention layer.

---

> ### Author Response · Authors · 2026-07-16
> **Response To Reviewer Pt7T (2 / 2)**
>
> These minimal post-training strategies when combined together yield a balance between confidence rating and response quality. Since the above three components are standard adoptions of an LM pipeline, our engineering contributions in OCC mainly boil down to the (1) choice of prefixes and (2) method of steering. Our empirical evaluation in Section 4 shows that prompt prefixes form the central part of our desiderata. Furthermore, our ablation studies and rating quality in Section 5 show that the choice of gumbel distribution for steering attention layers is found to be pivotal in minimizing overconfidence. These two additions and their usage in improving the confidence rating spread remain the sole contributions of OCC. Thus, the above two engineered aspects build on fundamental fixes (prompting and steering) with only  required modifications as their choice of implementation.
>
> > Have the authors compared this heavily engineered pipeline to much simpler calibration techniques, such as standard RLHF/DPO calibration, basic temperature scaling, or simple few-shot system prompting?
>
> Thank you for the suggestion. Following the reviewer’s feedback, we compare the checklist framework to three prompting and temperature scaling methods. Updated results and discussion can be found in Appendix D. We additionally present these below.
>
> Table below compares confidence rating and response quality of the OCC framework when compared to prompting and temperature scaling methods. Specifically, we compare with (1) *naive prompting* wherein participant model is given the questionnaire template, (2) *1-shot prompting* wherein the model is additionally given a completed questionnaire with responses corresponding to the same product, (3) *guided prompting* wherein the model is provided the average confidence rating for the given product questionnaire (using the following prompt - "Note that the average confidence for this questionnaire was <confidence>") and (4) *temperature scaling* wherein output logits $z$ of the model are scaled by a constant temperature parameter $T$, $\hat{z} = \text{softmax}(\frac{z}{T})$. In the case of Gemma 3 4B, we observe a saturated trend. OCC performance ratings are optimally minimal but saturate at 8.0 due to limited parameter budget. Response quality, compared using the judge model win rates against the base model response, demonstrate that a 4B parameter budget allows the model to stay performant with prompting and temperature scaling. In the case of Gemma 3 12B, we observe significant improvements in both minimizing overconfidence and response quality. a 12B parameter budget optimally balances between low confidence ratings ($14.87 \%$ improvement over guided prompting) and improved response quality over the base model. Thus, OCC is performant over prompting and temperature scaling methods utilizing prefixes, finetuning and steering.
>
> | Method | Confidence Rating ($\downarrow$) | GPT-4.1 Win Rate ($\uparrow$) | Claude 4 Win Rate ($\uparrow$) |
> |---|---:|---:|---:|
> | Gemma 3 4B (Naive Prompting) | 9.06 | 0.44 | 0.64 |
> | Gemma 3 4B (1-Shot Prompting) | 8.34 | 0.30 | 0.76 |
> | Gemma 3 4B (Guided Prompting) | 8.83 | 0.26 | **0.78** |
> | Gemma 3 4B (Temperature Scaling) | 8.78 | 0.40 | 0.66 |
> | Gemma 3 4B (OCC) | **8.08** | **0.46** | 0.50 |
> | Gemma 3 12B (Naive Prompting) | 8.79 | 0.30 | 0.78 |
> | Gemma 3 12B (1-Shot Prompting) | 8.48 | 0.42 | 0.80 |
> | Gemma 3 12B (Guided Prompting) | 9.14 | 0.42 | 0.60 |
> | Gemma 3 12B (Temperature Scaling) | 8.97 | 0.42 | 0.86 |
> | Gemma 3 12B (OCC) | **7.78** | **0.46** | **0.875** |
>
> We further compare relative activity and activation distributions of the first layer. Figure 12 presents the spread of relative activity for 4B and 12B participant models when using prompting and temperature scaling methods. In the case of 1-shot and guided prompting when compared to OCC, utilizing finetuning followed by steering pushes activations towards larger magnitude logits in the attention layers. This leads to a desirable spread wherein participant models attend and respond favorably to questionnaire prompts. Temperature scaling, on the other hand, presents a spread similar to OCC while not accounting for local changes in attention. Figure 13 presents the sum of logits above mean for 4B and 12B participant models. We observe that both prompting and temperature scaling methods present a similar spread as OCC but with higher variance. Intuitively, attention logits consistently attend and activate within the same local token range resulting in a robust activity spread in earlier layers.
>
> > On page 2, second line from the bottom: "ormpt intensity" should be "prompt intensity."
>
> Thank you, the grammatical error has now been corrected.
>
> Kindly let us know if our response above addressed your concerns. We would be happy to discuss and update the manuscript further.

---

### Review · Reviewer_Mun2 · 2026-07-04

**Summary Of Contributions:**

The paper focuses on the reliability of using language models (LMs) for subjective evaluations, particularly for focus group surveys on physical products, and seeks to address overconfidence of LM responses in such settings. Their proposed "Over-Confidence Checklist" (OCC) combines two techniques: (1) finetuning the model on responses generated by adding "cautionary" or "consequential" prefixes to the prompt, and (2) inference-time steering of model activations, specifically *Gumbel steering* along directions that are derived from training a sparse autoencoder (SAE) on a more powerful "teacher" model.

Across varying-size models from the Gemma family, they show that their method leads to a reduction in the model's self-reported confidence, give qualitative examples showing that the resulting responses contain fewer confabulated facts about the product, and show that their method achieves a higher win rate than the baseline (using GPT-4.1 and Claude 4 as judge models). Ablation studies investigate the role of different forms of steering, e.g., deterministic (instead of Gumbel) steering, steering of the residual stream instead of attention heads, showing that the proposed configuration outperforms the alternatives.

### Strengths (S)
1. **Significant, interesting problem:** The use of LMs for focus group surveys is important from two perspectives: as an actual use case, and as a way to evaluate LMs, with the paper focusing on the latter motivation. As expected, LMs behave differently than humans, and the general goal of making LMs behave more similar to humans has many downstream applications. The paper specifically focuses on overconfidence of LMs, which is a hard problem from a conceptual standpoint: how do we know the "correct" level of confidence? Although this point is not specifically addressed (the authors instead prescribe a confidence level, without discussing any notion of "correctness"), the paper raises an interesting conversation.
2. **Combination of techniques:** The paper does not stop at finetuning, but also uses steering guidance in an interesting way. The steering technique is fairly interesting, combining a teacher model, sparse autoencoders, and randomization (via a Gumbel distribution).
### Weaknesses (W)
1. **Mathematical/technical clarity:** More of the mathematical terms, especially those important to the method and empirical findings, should be properly defined. Some of these are basic and can be inferred from the description, but adding an equation would increase clarity, e.g. "sum of activations above mean value". In other places, the text leaves too much ambiguity and an equation is necessary for clarity:
	1. "We denote $\mathcal{T}(h_i)$ as the operation of the SAE encoder and selecting the top-$k$ logits from its latent features": The notation doesn't specify $k$, and the latent features of an SAE are not logits.
	2. In Table 1, how is entropy computed/estimated?
	3. For the steering, "We thus select the Gumbel distribution... over hidden states $h_i$ as a natural statistical choice and steer the first layer", I could not tell how dimensionality/alignment issues are handled. The teacher model activations $h_i$ may be of a higher dimension than the student model's first-layer activations, and even if they are the same dimensionality, the two models may not be aligned (e.g. they may differ by an unknown linear transformation).
2. **Motivation for finetuning vs. prefixing:** As a way of modifying a LM's behavior, finetuning is relatively expensive when compared to inference-time techniques like steering, or simple prompt engineering. How do the results compared when just including the prompt prefixes during inference, without finetuning? I can imagine explanations for why finetuning is necessary, but did not feel that enough motivation was given.
3. **Possibly cherry-picked thresholds:** It's not clear that the method would work for arbitrary thresholds $\beta_1$ and $\beta_2$: beside the if-statements in the checklist, the thresholds aren't used in the method. After finetuning on infilled responses, the model could still be overconfident compared to the prescribed thresholds. I would have expected some kind of iteration until the thresholds are met. Instead, the experiments "pick" confidence thresholds (and activation thresholds) that are suspiciously close to the achieved value (e.g. for the second case, the activation threshold is $\eta = 1.35$ and the achieved value is 1.3566). The experiments should be run with multiple thresholds, which may require changing the method.
### Minor Weakness (MW)
1. **Structure of related work:** The related work reads like a list, without much connection between each work or between these works and the paper. In "Overconfidence in Language Models", the section alternates between works that are focused on understanding/uncovering overconfidence, and works that are focused on mitigating overconfidence, with a wide variety of mitigation strategies. Each work does not need its own sentence, they can be grouped together by theme, and there should always be a connection back to the present work.
2. **Missing logical connections:** As motivation, the work deeply relies on a (negative) correlation between (1) the amount of activity in early layers and (2) the model's internal degree of confidence. While this connection is intuitively reasonable (higher activity $\approx$ more explanations competing for dominance $\approx$ lower internal confidence), it doesn't seem like a logical necessity, but a hypothesis that needs empirical validation. Was this connection established in prior work? If so, I missed the reference, in which case it probably needs more emphasis. Or, is this connection meant as a novel insight? The sentence "We now relate the spread of activations with confidence ratings" seems to suggest that the authors will establish this connection, but Figure 5 only plots model size vs. confidence, there does not seem to be a plot of early layer activity vs. confidence.
3. **Grammar, terminology, and citation format:** There are some minor issues that could have been caught by passing through a simple tool (e.g. Grammarly, ChatGPT, Claude):
	1. *Grammar:* Many sentences need to start with "the", e.g. on page 8, "Model assigns a confidence rating" should be "The model assigns a confidence rating". This grammatical issue is prevalent throughout (probably ~10-20 times).
	2. *Terminology:* "Desiderata" means "things that are desired" from a method; in several places, it is used to refer to the method itself (e.g. "following our desiderata, we elicit survey ratings"). Further, the proposed method is referred to as a "checklist", which is very atypical usage: just refer to it as a method, and write it out as standard pseudocode (which is essentially the role of Figure 1).
	3. *Citation format:* In-text citations should not be parenthetical; this issue is prevalent throughout the related work (Section 2).
4. **Figures/tables:** The text should be larger in some figures (e.g., Figure 10). The caption for Figure 4 includes "(bottom)", but the figure is one row. In Table 3, the value "0.55" is bolded in the Gemma 3-12B / Claude 4 column, which is not the highest value.

**Audience:**

Yes

**Audience Explanation:**

Modifying the behavior of LMs is one of a very timely, important, and rich topic in machine learning, and an LM's self-reported confidence is a crucial aspect of the behavior that deserves study.

**Broader Impact Concerns:**

Broader impacts issues are already addressed.

**Claims And Evidence:**

No

**Claims Explanation:**

The main claims are that LM survey participants are confident in their responses, and that their over-confidence checklist (OCC) approach reduces overconfidence. The main problem in supporting these claims is the difficulty of rigorous evaluation: there is no "ground truth" about the correct level of confidence. The plots do show that LMs have a high degree of confidence, and that their method reduces the LM's self-reported confidence, but the claims about *over*-confidence require more rigorous experimental design / evaluation methodology.

**Requested Changes:**

1. **Increase mathematical clarity:** See W1. (Critical for securing recommendation for acceptance)
2. **Add motivation for finetuning over prompt modification:** See W2. (Would simply strengthen the work)
3. **Add experiments showing how the method achieves different prescribed thresholds:** See W2. (Critical for securing recommendation for acceptance)
4. **Make related work more connected:** See MW1. (Would simply strengthen the work)
5. **Fix other minor issues:** See MW2-4. (Would simply strengthen the work)

---

> ### Author Response · Authors · 2026-07-16
> **Response To Reviewer Mun2 (1 / 4)**
>
> We thank the reviewer for providing feedback which is of utmost value to our work. Below we aim to address your concerns with changes in the revision marked in $\color{blue}{\text{blue}}$.
>
> > Mathematical/technical clarity: More of the mathematical terms, especially those important to the method and empirical findings, should be properly defined. Some of these are basic and can be inferred from the description, but adding an equation would increase clarity, e.g. "sum of activations above mean value".
>
> Thank you for raising your concern, following your suggestions we have added equations and made our mathematical formulations more clear. Specifically, we explicitly add an equation and its description for the sum of activations above mean value in Section 5. Below is the updated description -
>
> We monitor the sum of activations above mean value $\hat{\eta}$ presented in Equation below. Given an activation vector $h\_{L}^{T}$ at layer $L$ for token $T$ of dimension $D$ in $\mathbb{R}^{D}$, we first compute its mean $\bar{h}\_{L}^{T}$ and sum all entries $h\_{L}^{T}(i)$ above the mean value.
>
> $\hat{\eta} = \sum\_{i = 1}^{D} h\_{L}^{T}(i)\cdot \mathbf{1}[h\_{L}^{T}(i) > \bar{h}\_{L}^{T}]; \quad \quad \bar{h}\_{L}^{T} = \frac{\sum_{i = 1}^{D} h_{L}^{T}(i)}{D}$
>
> > "We denote  as the operation of the SAE encoder and selecting the top- logits from its latent features": The notation doesn't specify , and the latent features of an SAE are not logits.
>
> We have modified the notation to incorporate k, $\mathcal{T}\_{k}(h_{i})$ and added corresponding equations and descriptions for both the SAE encoder and TopK operations. We further segregate the SAE training objective as a dedicated equation for more clarity. SAE latent features are now referred to as entries of the latent feature.
>
> > In Table 1, how is entropy computed/estimated?
>
> For our motivational experiments, we compute entropy and skewness for each activation vector and average across all samples. Specifically, given an activation vector $h_{L}^{T}$ at layer $L$ and token $T$ of dimension $D$ in $\mathbb{R}^{D}$, we compute the entropy and skewness of $h_{L}^{T}$. These metrics are then averaged across all prompts in the test set. In the case of first two layers, L is varied from 1 to 2. We have added this discussion to Appendix B.
>
> > I could not tell how dimensionality/alignment issues are handled. The teacher model activations  may be of a higher dimension than the student model's first-layer activations, and even if they are the same dimensionality, the two models may not be aligned (e.g. they may differ by an unknown linear transformation).
>
> Thank you for raising your concern. In case of a dimensional mismatch, the hidden state $h_{i}$ is compressed either using PCA or the encoder of a pretrained SAE. We primarily use the SAE encoder as it provides richer features while preserving the structure and dimensionality of hidden state vector. We have made the handling of dimensionality mismatch explicit in the main text.
>
> In the case of same dimensionality and a lack of alignment, we emphasize that the steering vector is sampled proportionately to the target model’s activations. This is controlled via steering strength. Specifically, we compute cosine similarities between the teacher model’s embeddings and target model’s activations. Embeddings that present high similarity are utilized for sampling the steering vector. This ensures that teacher embeddings do not inject noise in the activation space of the current target model.
>
> > Motivation for finetuning vs. prefixing: As a way of modifying a LM's behavior, finetuning is relatively expensive when compared to inference-time techniques like steering, or simple prompt engineering. How do the results compared when just including the prompt prefixes during inference, without finetuning? I can imagine explanations for why finetuning is necessary, but did not feel that enough motivation was given.
>
> Following the reviewer’s concern, we have added additional results in the form of comparisons and an ablation in Appendix D.

---

> > ### Author Response · Authors · 2026-07-16
> > **Response To Reviewer Mun2 (2 / 4)**
> >
> > Firstly, Table 5 compares confidence rating and response quality of the OCC framework when compared to prompting and temperature scaling methods. Specifically, we compare with (1) *naive prompting* wherein participant model is given the questionnaire template, (2) *1-shot prompting* wherein the model is additionally given a completed questionnaire with responses corresponding to the same product, (3) *guided prompting* wherein the model is provided the average confidence rating for the given product questionnaire (using the following prompt - "Note that the average confidence for this questionnaire was <confidence>") and (4) *temperature scaling* wherein output logits $z$ of the model are scaled by a constant temperature parameter $T$, $\hat{z} = \text{softmax}(\frac{z}{T})$. In the case of Gemma 3 4B, we observe a saturated trend. OCC performance ratings are optimally minimal but saturate at 8.0 due to limited parameter budget. Response quality, compared using the judge model win rates against the base model response, demonstrate that a 4B parameter budget allows the model to stay performant with prompting and temperature scaling. In the case of Gemma 3 12B, we observe significant improvements in both minimizing overconfidence and response quality. a 12B parameter budget optimally balances between low confidence ratings ($14.87 \%$ improvement over guided prompting) and improved response quality over the base model. Thus, OCC is performant over prompting and temperature scaling methods utilizing prefixes, finetuning and steering.
> >
> >
> > Secondly, we specifically ablate finetuning for prompting using our prompt prefixes of varying prefix intensities. Figure 24 presents the comparison of finetuning with prompting for different prefixes. In the case of prompting, participant models are not finetuned and only prompted using respective prefix types (standard, cautionary and consequential). The comparison reveals a common trend, across prefix types, finetuning on template questionnaires with set prefixes better influences the confidence rating when compared to naive prompting with the same set prefix. Intuitively, finetuning trades off internal confidence for computational expense by guiding activation patterns towards conservative rating behaviors.
> >
> > > Possibly cherry-picked thresholds: It's not clear that the method would work for arbitrary thresholds  and : beside the if-statements in the checklist, the thresholds aren't used in the method. After finetuning on infilled responses, the model could still be overconfident compared to the prescribed thresholds. I would have expected some kind of iteration until the thresholds are met. Instead, the experiments "pick" confidence thresholds (and activation thresholds) that are suspiciously close to the achieved value (e.g. for the second case, the activation threshold is  and the achieved value is 1.3566). The experiments should be run with multiple thresholds, which may require changing the method.
> >
> > We recall that confidence thresholds are manually selected by the user to calibrate for a desirable level of confidence. This is because subjective evaluations lack a ground-truth and are customizable as per the behavior of an LM. A user must set confidence thresholds and follow the OCC to obtain a desirable level of confidence ratings. Similarly, a user sets the activation threshold and follows the OCC. We emphasize that while confidence ratings must reduce following either cautionary or consequential prefixes, we never claim that activation sums will always drop below the specified threshold. This is because activation magnitudes are relative and vary for each model. Smaller models observe larger swings in their activation values while larger models present limited variation. However, our analysis in Section 5 and Appendix D shows that an increase in activation magnitudes is the direct cause of reduced confidence ratings. We have clarified this remark in the main text of Section 5.
> >
> > While we do not cherry pick confidence thresholds, we agree that thresholds require careful selection before following the OCC. That is, too high or low values of $\beta\_{1}$ and $\beta\_{2}$ will require adjusting samples for finetuning and modulating the steering weight during inference. If threshold values are too low, finetuning and steering budget increases. On the other hand, if thresholds are too high finetuning and steering budget decreases. We elucidate this with additional experiments in Appendix D and below for two more thresholds.

---

> > > ### Author Response · Authors · 2026-07-16
> > > **Response To Reviewer Mun2 (3 / 4)**
> > >
> > > We note that while confidence thresholds may be chosen arbitrarily, obtaining desirable confidence ratings also depends on hyperparmeter tuning. The Table below presents the change in confidence for Gemma 3 models across two different randomly threshold configurations. In the case of lower thresholds, we require a larger finetuning set of samples and a greater steering strength during inference. This allows activations to gradually adapt while trading off reliability for post-training budget. In the case of higher thresholds, lesser finetuning and steering effort is required allowing model activations to saturate with higher internal beliefs. As our checklist employs finetuning and steering as key ingredients for confidence alignment, we emphasize that their hyperparemter tuning play a central role in obtaining appropriate ratings (similar to custom modern-day language inference tasks and protocols).
> > >
> > > | Participant Model | $\(\beta_1\)$ | $\(\beta_2\)$ | Confidence Before | Confidence After |
> > > |---|---:|---:|---:|---:|
> > > | Gemma 3 4B | 9.0 | 7.0 | 9.4 | 8.01 |
> > > | Gemma 3 12B | 9.0 | 7.0 | 8.9 | 7.00 |
> > > | Gemma 3 27B | 9.0 | 7.0 | 8.8 | 6.7 |
> > > | Gemma 3 4B | 9.5 | 8.0 | 10.0 | 8.0 |
> > > | Gemma 3 12B | 9.5 | 8.0 | 8.0 | 7.0 |
> > > | Gemma 3 27B | 9.5 | 8.0 | 9.0 | 7.0 |
> > >
> > > > Structure of related work: The related work reads like a list, without much connection between each work or between these works and the paper. In "Overconfidence in Language Models", the section alternates between works that are focused on understanding/uncovering overconfidence, and works that are focused on mitigating overconfidence, with a wide variety of mitigation strategies. Each work does not need its own sentence, they can be grouped together by theme, and there should always be a connection back to the present work.
> > >
> > > Thank you for raising your concern. Following your suggestions, we have restructured the related work section of the main text. We implement two changes, (1) citations are grouped and discussion is made more concise, (2) discussions link back to common contributions, central themes and original pieces of text which is the present work.
> > >
> > > > Missing logical connections: As motivation, the work deeply relies on a (negative) correlation between (1) the amount of activity in early layers and (2) the model's internal degree of confidence. While this connection is intuitively reasonable (higher activity  more explanations competing for dominance  lower internal confidence), it doesn't seem like a logical necessity, but a hypothesis that needs empirical validation. Was this connection established in prior work? If so, I missed the reference, in which case it probably needs more emphasis. Or, is this connection meant as a novel insight? The sentence "We now relate the spread of activations with confidence ratings" seems to suggest that the authors will establish this connection, but Figure 5 only plots model size vs. confidence, there does not seem to be a plot of early layer activity vs. confidence.
> > >
> > > Once again, we conduct additional experiments and provide empirical evidence that links higher activity in layers of the model with its confidence ratings. We compare the relationship between activity in layers of a model with its internal confidence rating across all prefix types. Figure 25 presents the variation of L1, L2 and Infinity norms of activations of all layers and first layer with their respective confidence ratings. We make two observations. Firstly, lowering confidence ratings arises as a consequence of growing model activity indicating that confidence is internalized within the structure and magnitude of activation entries. Earlier layers drive model activity towards larger scales allowing layers to explain and attend to prefixed templates in entirety, hence resulting in balanced confidence ratings. Secondly, highly confident models are the ones with lower parameter budgets (Gemma 3 4B) irrespective of prefix types. This validates our previous observations that in addition to steering and finetuning, a larger parameter budget also aids in mitigating highly confident responses.
> > >
> > > > Grammar: Many sentences need to start with "the", e.g. on page 8, "Model assigns a confidence rating" should be "The model assigns a confidence rating". This grammatical issue is prevalent throughout (probably ~10-20 times).
> > >
> > > Thank you, we have corrected grammatical errors and appended “the” in appropriate locations (primarily in Sections 4 and 5).
> > >
> > > > Terminology: "Desiderata" means "things that are desired" from a method; in several places, it is used to refer to the method itself (e.g. "following our desiderata, we elicit survey ratings"). Further, the proposed method is referred to as a "checklist", which is very atypical usage: just refer to it as a method, and write it out as standard pseudocode (which is essentially the role of Figure 1).

---

> > > > ### Author Response · Authors · 2026-07-16
> > > > **Response To Reviewer Mun2 (4 / 4)**
> > > >
> > > > We have rectified our terminology and usage of desiderata. We use desiderata primarily in Section 4 to refer to qualities of long-term awareness and rating diversity. For other cases in Section 5, we simply refer to our method.
> > > >
> > > > We refer to our framework in a checklist format, as opposed to an algorithm, primarily due to its asynchronous and step-by-step approach. We note that the steps of eliciting ratings, inspecting activations, assessing confidence thresholds and activation sums are items on a list, and not simply steps in an algorithm. This is because as per an algorithm’s if condition, if confidence threshold falls below $\beta_{1}$ but remains above $\beta_{2}$, it will fall in the second if block and execute infilling/finetuning on cautionary prefixes. This may further reduce confidence ratings making the model underconfident, and hence deteriorating judgement quality. However, our nomenclature of a checklist forces the practitioner to select one of the two if blocks, and not both. This prevents a blowup of model activity (as activity grows with reducing confidence ratings) while maintaining judgement quality and ratings in a desirable custom range as specified by the user.
> > > >
> > > > > Citation format: In-text citations should not be parenthetical; this issue is prevalent throughout the related work (Section 2).
> > > >
> > > > Thank you, we have removed parenthesis in our citations.
> > > >
> > > > > Figures/tables: The text should be larger in some figures (e.g., Figure 10). The caption for Figure 4 includes "(bottom)", but the figure is one row. In Table 3, the value "0.55" is bolded in the Gemma 3-12B / Claude 4 column, which is not the highest value.
> > > >
> > > > Thank you, we have corrected the caption of Figure 4 and Table 3 entry.
> > > >
> > > > Kindly let us know if our response above addressed your concerns. We will be happy to further discuss and update the manuscript following your suggestions.

---

> > > > > ### Comment · Reviewer_Mun2 · 2026-07-17
> > > > >
> > > > > I thank the authors for their thorough response, which addressed some of my main concerns. Here's a brief summary of each point, including whether I'm satisfied with the response, and any additional questions, suggestions, or concerns.
> > > > >
> > > > > W1. **Mathematical/technical clarity:** I'm happy with the definitions of $\hat{\eta}$ and the updated notation/language around the SAE encoder. I'm still not satisfied with [W1.2] or [W1.3]:
> > > > >     - Entropy is defined with respect to a distribution $p$ as the expectation of $\log p(x)$ with respect to $p$. You need to specify how you relate $h^T_L$ to a distribution (e.g., do you fit a normal distribution?) and thus an entropy.
> > > > >     - The explanation in the response is reasonable, but the paper needs more detail; the authors have only added a mention of the factor that $h_i$ is compressed using PCA or the encoder of an SAE, and did not mention using cosine similarities for alignment. These steps need to be explicitly described.
> > > > >
> > > > > W2. **Motivation for finetuning vs. prefixing:** My understanding (from the new Figure 24) is that finetuning has a larger effect on confidence reduction, which is why it is preferred over just prompting. This effect is interesting, but the one figure does not fully satisfy my concerns. First, it's important to compare other metrics like win rate, to get a full picture of the improvement/tradeoffs. Second, the paper still needs to better motivate finetuning over prompting - this motivation should be explained in the introduction to justify the additional computational expense.
> > > > >
> > > > > W3. **Thresholds:** My understanding is that, if the method does not achieved the desired confidence rates set by the user, then the user will need to re-run the method with a new activation threshold and steering strength, possibly several times until it achieves the desired confidence rates. As the authors say, *"while confidence thresholds may be chosen arbitrarily, obtaining desirable confidence ratings also depends on hyperparameter tuning."* To me, this is an acceptable limitation, but one that should be emphasized, and the authors should give more guidance on hyperparameter selection. For example, with the different choices of $\beta_1$/$\beta_2$ given in the response, the authors should also report the hyperparameters, so users can get a sense of the underlying scaling.
> > > > >
> > > > > MW1. **Related work:** I'm happy with the new related work section.
> > > > >
> > > > > MW2. **Logical connections:** Thank you for adding Figure 25, which directly compares activation strengths and average confidence. There does appear to be a weak negative correlation for standard prompts, though I would be careful about using causal language. Since the method is directly motivated by this phenomenon, this finding should be referenced early on.
> > > > >
> > > > > MW3. **Grammar/terminology/citation format:** I'm mostly happy with the changes to grammar and terminology, and appreciate the revisions. However, I still think "algorithm" is better than "checklist", and would result in more clarity: the inputs are a language model $f_\theta$, hyperparameters $\beta_1, \beta_2, \eta$ (as well as one for steering strength), a set of consequential prefixes, a set of cautionary prefixes, and a teacher model, and the output is a finetuned model. The author's response indicates a somewhat strange understanding of algorithms: an algorithm *is* a step-by-step approach, and the comment about "select on of the two if blocks, and not both" can be handled by an if-else or case statement.
> > > > >
> > > > > MW4. **Figures/tables:** Thanks for the updates!

---

> > > > > > ### Author Response · Authors · 2026-07-18
> > > > > > **Follow Up To Response**
> > > > > >
> > > > > > Thank you for promptly following up on our rebuttal. We provide our response below.
> > > > > >
> > > > > > > You need to specify how you relate to a distribution (e.g., do you fit a normal distribution?) and thus an entropy. - The explanation in the response is reasonable, but the paper needs more detail;
> > > > > >
> > > > > > Yes, given $h_{L}^{T}$ as a D-dimensional array, we fit a Gaussian distribution whose mean and variance are that of $h_{L}^{T}$. Following your concern, we have made this explicit in Appendix B.
> > > > > >
> > > > > > > the authors have only added a mention of the factor that  is compressed using PCA or the encoder of an SAE, and did not mention using cosine similarities for alignment. These steps need to be explicitly described.
> > > > > >
> > > > > > We apologize for the incompletion. The discussion on Page 6 now contains the full description including utilization of cosine similarity. Additionally, in Appendix B we have added a detailed step-by-step process of how the steering vector is selected for a set of target activations.
> > > > > >
> > > > > > > First, it's important to compare other metrics like win rate, to get a full picture of the improvement/tradeoffs. Second, the paper still needs to better motivate finetuning over prompting - this motivation should be explained in the introduction to justify the additional computational expense.
> > > > > >
> > > > > > Thank you for raising your concern. Table 2 describes this comparison between a range of prompting and temperature scaling schemes and finetuned models. We observe that finetuning is found to be beneficial in both improving response quality (via judge win rates) and internal beliefs (via reducing confidence ratings). For a more direct comparison, we compare win rates using Claude Sonnet 4 as a judge in Table 4. Here, steering is disabled and finetuning was conducted using the same prompts used for naive prompting. We consistently observe that judge preferences taper towards finetuned responses. Intuitively, finetuning distills truthful responses from frontier models which are necessary for improving both diversity and response quality. We have added this additional discussion in Section 1.
> > > > > >
> > > > > > > To me, this is an acceptable limitation, but one that should be emphasized, and the authors should give more guidance on hyperparameter selection. For example, with the different choices of / given in the response, the authors should also report the hyperparameters, so users can get a sense of the underlying scaling.
> > > > > >
> > > > > > Thank you for following up on the results. We have added the corresponding hyperparameters in Table 3, mainly the number of finetuning samples and the normalized steering strength. Additionally, we provide a general guide for hyperparameter tuning in Appendix B with the below discussion. We have also added discussion in Appendix D emphasizing the role of hyperparameter tuning for different confidence thresholds.
> > > > > >
> > > > > > We note that thresholds require careful selection before following the algorithm. That is, too high or low values of thresholds will require adjusting samples for finetuning and modulating the steering weight during inference. If threshold values are too low, finetuning and steering budget increases. On the other hand, if thresholds are too high, finetuning and steering budget decreases. A general rule of thumb is to finetune with 50 samples and steer with strength 1 for higher thresholds. For lower thresholds, these values must be increased proportional to the desired confidence rating with more weight given to finetuning samples.
> > > > > >
> > > > > >
> > > > > > > There does appear to be a weak negative correlation for standard prompts, though I would be careful about using causal language. Since the method is directly motivated by this phenomenon, this finding should be referenced early on.
> > > > > >
> > > > > > Following your concern, we provide additional motivation in Section 1. Additionally, in section 4 we discuss the phenomenon and direct the reader to Appendix for the result. If the reviewer prefers, in our final version we aim to present this result in the main text by reallocating space and shifting some of our distributional plots to the Appendix.
> > > > > >
> > > > > > > However, I still think "algorithm" is better than "checklist", and would result in more clarity: the inputs are a language model , hyperparameters  (as well as one for steering strength), a set of consequential prefixes, a set of cautionary prefixes, and a teacher model, and the output is a finetuned model.
> > > > > >
> > > > > > Thank you. Following your suggestion, we have renamed the scheme to Confidence Tuning Algorithm (CTA) and added the if-else statement to simplify case operations. We believe this terminology is more appropriate with the workflow of an algorithm and also aligns with the objective of customizing confidence ratings, as opposed to aggressively minimizing *over*-confidence against absent ground truths.
> > > > > >
> > > > > > Once again, thank you for constructive comments and suggestions. Please let us know if any concerns remain and we will try our best to address them.

---

### Review · Reviewer_k8oV · 2026-07-05

**Summary Of Contributions:**

This paper investigates the issue of overconfidence in Large Language Models (LLMs) when deployed as participants in subjective focus group surveys. By analyzing the early transformer layers, the authors identify key correlations between prompt intensity, internal activation patterns, and self-reported confidence scores. To mitigate this, they introduce the Over-Confidence Checklist (OCC), a mitigation framework that integrates prompt-based fine-tuning with activation steering via teacher hidden states, Gumbel sampling, and sparse autoencoders. Evaluated on Gemma 3 models, the OCC successfully curbs overconfident outputs and elevates the overall quality of subjective survey responses.

**Audience:**

Yes

**Audience Explanation:**

Investigating LLM overconfidence remains a highly active and significant avenue of research.

**Claims And Evidence:**

No

**Claims Explanation:**

The current evidence does not sufficiently substantiate the paper's primary claims due to several critical gaps:

- Weak Link Between Analysis and Methodology: The transition from the empirical findings to the proposed solution is poorly justified. While the paper outlines various activation statistics (such as entropy, skewness, and top-K activations), it fails to explain how these specific insights directly inform the design of the methodology.

- Insufficient Component Motivation: Key architectural choices—specifically the integration of Gumbel sampling, sparse autoencoders, and top-K activation metrics—lack clear theoretical or empirical justification.

- Absence of Rigorous Ablation Studies: The necessity of each individual component within the framework is not adequately demonstrated. A more comprehensive ablation analysis is required to isolate and validate the performance contribution of each module.

- Correlation vs. Causation: The authors frequently derive major conclusions from purely descriptive empirical observations, such as shifted activation distributions, without proving a definitive causal link to model overconfidence.

Furthermore, the paper's methodological novelty is poorly defined, obscured by a lack of clarity regarding what the experiments are intended to validate. This ambiguity is compounded by the omission of critical implementation details—such as the exact mathematical definition of the top-$K$ activation statistic—which ultimately hinders the interpretability and reproducibility of the experimental results.

**Requested Changes:**

- Explicating Methodological Contribution: The authors should explicitly define the proposed algorithm, clearly delineating which components are novel and distinguishing the framework from established baselines in prompt engineering, activation steering, and sparse autoencoders (SAEs).

- Strengthening Design Justifications: Provide a more rigorous rationale for specific architectural choices—such as Gumbel sampling, SAE-based steering, and top-$K$ activation statistics—backed by theoretical grounding, empirical data, or comparisons against intuitive baselines.

- Bridging Analysis and Methodology: The manuscript needs to demonstrate a causal link between the observed activation patterns and model overconfidence.

- Expanding Ablation Frameworks: Conduct thorough ablation studies that isolate each individual component (e.g., prompt-based fine-tuning, activation steering, Gumbel sampling, and SAEs) to empirically justify their inclusion in the pipeline.

- Ensuring Reproducibility and Detail: Clarify critical implementation variables. Specifically, provide a precise mathematical definition for metrics like the top-$K$ activation statistics (including the exact value of $K$ and its calculation method) to ensure the work can be accurately interpreted.

---

> ### Author Response · Authors · 2026-07-16
> **Response To Reviewer k8oV (1 / 3)**
>
> Thank you for your review. Below are our responses to your concerns with changes in the revision marked in $\color{blue}{\text{blue}}$.
>
> > Weak Link Between Analysis and Methodology: The transition from the empirical findings to the proposed solution is poorly justified. While the paper outlines various activation statistics (such as entropy, skewness, and top-K activations), it fails to explain how these specific insights directly inform the design of the methodology.
>
> We recall that OCC is motivated by following the desiderata of (1) long-term awareness and (2) rating diversity, both of which are introduced and described in Section 4. In Section 5, we further describe that introducing these desirable traits in a participant model requires external changes in model queries and changes in model activations. This discussion and motivation is provided in Section 5 paragraph 1 wherein we highlight that (1) modulating the intensity of instructions and (2) minimizing deterministic behaviors serves as the key motivating factors for the design of the checklist. Below we present this discussion (verbatim) for your visibility.
>
> Firstly, introducing long-term awareness requires external changes in model queries. Naturally, modulating the intensity of instructions forces participants to attend towards uncertainty in their responses. Similarly, minimizing deterministic rating behaviors requires internal changes in model activations. Specifically, reshaping the activation distribution via a learned prior induces diversity and guides participants towards aligned behaviors. These external and internal changes, combined with appropriate parameter budget and choice of interventions, form the basis for addressing overconfidence.
>
>
> > Insufficient Component Motivation: Key architectural choices—specifically the integration of Gumbel sampling, sparse autoencoders, and top-K activation metrics—lack clear theoretical or empirical justification.
>
> The reviewer claims insufficient component motivation. We remind the reviewer that these components are adequately introduced and motivated in detail in Section 4 and 5. Below we provide their respective locations and motivations-
>
> Top-K activation statistic - Our top-k activation statistic, that is the sum of activations above mean value, is introduced and motivated mathematically in detail in Section 5 (paragraphs 2 and 3). We emphasize that the quantity represents an approximation of top-k activations by selecting the ones above the mean quantile.
> Gumbel sampling - Gumbel sampling of the steering vector is motivated from a statistical perspective in Section 5 (paragraph 4). Specifically, we utilize a teacher model as a learned prior that embeds prefixed prompts and responses into its hidden states. The hidden state serves as a sample from an expressive distribution unbiased in its preferences and rating choices. Steering vector z is then sampled from the Gumbel distribution over teacher hidden states and added to activations using weighted steering. Utilizing the gumbel distribution as a steering prior provides two key benefits. Firstly, the distribution, representing the distributions of extreme activation values, yields a concentrated spread with shorter tails implicitly increasing certainty and eliminating excess activity. Secondly, sampling a steering vector reallocates activation budget towards inactive neurons by trading off stability for diversity.
> SAEs - SAEs are theoretically introduced and motivated in Section 3. Their usage is further explained in the discussion in Section 5. In order to omit activation noise, we train a SAE on teacher hidden states and sample using its encoder. Enforcing a sparsity constraint minimizes logit noise while yielding disentangled features for steering.

---

> ### Author Response · Authors · 2026-07-16
> **Response To Reviewer k8oV (2 / 3)**
>
> > Absence of Rigorous Ablation Studies: The necessity of each individual component within the framework is not adequately demonstrated. A more comprehensive ablation analysis is required to isolate and validate the performance contribution of each module.
>
> The reviewer claims the absence of rigorous ablation studies. We respectfully remind the reviewer that our experiments conduct leave-one-out ablations for gumbe sampling, prefix types, steering locations and steering types in Section 5 and Appendix D. We lay an emphasis on these as steering and prefix intensity form the main contributions of the OCC. Further, we have conducted additional experiments and provide ablations for prompting variations and finetuning. Below is a list of ablations (and their respective locations in the manuscript) -
> * Gumbel Steering - Figure 10 and Figure 19 ablates the choice of gumbel distribution against naive steering and demonstrates that heavy-tailed gumbel distribution is found as a key component in reducing confidence ratings. Furthermore, Table 3 compares ablates gumbel steering for both cautionary and consequential prefixes and presents suitable response quality via judge preferences.
> * Steering Location - Figure 18 compares steering at layer 1, layer 15 and layer 30 by analyzing the structure of altered activation distributions. We observe that steering in earlier and middle layers is found to better stabilize activity internally and more directly influence confidence ratings.
> * Steering Type - Figure 10 compares attention steering with residual steering and demonstrates that steering attention heads is found to be beneficial in reducing overconfidence.
> * Prefix Intensity - Figure 3, Table and Figure 19 compare the role of varying prefix intensity (standard, cautionary and consequential prefixes) both with and without gumbel steering. We observe that consequential prefixes are most performant in minimizing confidence ratings followed by cautionary prefixes which inform participants of long-term response consequences thereby inducing response awareness.
> * Prompting and Temperature Scaling - Table 5 compares confidence rating and response quality when compared to prompting and temperature scaling methods. Specifically, we compare with (1) *naive prompting* wherein participant model is given the questionnaire template, (2) *1-shot prompting* wherein the model is additionally given a completed questionnaire with responses corresponding to the same product, (3) *guided prompting* wherein the model is provided the average confidence rating for the given product questionnaire (using the following prompt - "Note that the average confidence for this questionnaire was <confidence>") and (4) *temperature scaling* wherein output logits $z$ of the model are scaled by a constant temperature parameter $T$, $\hat{z} = \text{softmax}(\frac{z}{T})$. In the case of Gemma 3 4B, we observe a saturated trend. OCC performance ratings are optimally minimal but saturate at 8.0 due to limited parameter budget. Response quality, compared using the judge model win rates against the base model response, demonstrate that a 4B parameter budget allows the model to stay performant with prompting and temperature scaling. In the case of Gemma 3 12B, we observe significant improvements in both minimizing overconfidence and response quality. a 12B parameter budget optimally balances between low confidence ratings ($14.87 \%$ improvement over guided prompting) and improved response quality over the base model. Thus, OCC is performant over prompting and temperature scaling methods utilizing prefixes, finetuning and steering.
> * Finetuning - We specifically ablate finetuning for prompting using our prompt prefixes of varying prefix intensities. Figure 24 presents the comparison of finetuning with prompting for different prefixes. In the case of prompting, participant models are not finetuned and only prompted using respective prefix types (standard, cautionary and consequential). The comparison reveals a common trend, across prefix types, finetuning on template questionnaires with set prefixes better influences the confidence rating when compared to naive prompting with the same set prefix. Intuitively, finetuning trades off internal confidence for computational expense by guiding activation patterns towards conservative rating behaviors.

---

> > ### Author Response · Authors · 2026-07-16
> > **Response To Reviewer k8oV (3 / 3)**
> >
> > > Correlation vs. Causation: The authors frequently derive major conclusions from purely descriptive empirical observations, such as shifted activation distributions, without proving a definitive causal link to model overconfidence.
> >
> > Our analysis relies on the distributional nature of activations. Activation distributions demonstrate visible shifts in activity pattern and orientation when steered or prompted with prefixes of varying intensity. This is evident in Figures 2, 4 and 10 wherein distributions deviate from their central mean values and demonstrate multi-modal nature. While distributions drive the empirical evidence and reason for variation, a direct causal link is established using quantitative results in Tables 1, 2 and 3 and Figures 3, 5, 6 and 10. We observe a direct link between model overconfidence and sum of activations above mean wherein an increase in the statistic leads to a reduction in highly confident ratings across different parameter budgets. Furthermore, we observe a definitive link between varying prefix intensity and reduction in overconfidence in Figure 3 and Figure 7 wherein increasing prefix intensity proportional to confidence ratings is found to induce a conservative rating pattern.
> >
> > Additionally, we argue that the distributional perspective is informative not only in providing a qualitative verification, but it also demonstrates a visible shift in model behavior. We only follow the line of prior established works which link qualitative model behavior with quantitative empirical evidence [1, 2, 3, 4].
> >
> >
> > > This ambiguity is compounded by the omission of critical implementation details—such as the exact mathematical definition of the top- activation statistic—which ultimately hinders the interpretability and reproducibility of the experimental results.
> >
> > The reviewer claims omission of implementation details and definitions. We remind the reviewer that Section 3 (Preliminaries) holistically consists of mathematical notations and definitions which are utilized within the work. Activation statistics such as entropy and skewness are introduced in Table 1 and the paragraph below it. The primary Top-k activation statistic, sum of activations above mean value, is also introduced in Section 5 (paragraph 2). We have added an exact mathematical expression for the sum of activations above mean value and clarified it as our main statistic.
> >
> > Further, we emphasize that implementation details, dataset descriptions and additional reproducibility notes can be found in Appendix B and Appendix C. These provide detailed reproducibility guidelines and our experimental setup from an implementation perspective including the exact value of K = 200.
> >
> >
> > We kindly request the reviewer to review the revised changes and let us know if our response above addressed the concerns.
> >
> > [1]. Holtzman et al, The Curious Case Of Neural Text Degeneration, ICLR 2020.
> > [2]. Finlayson et al, Closing the Curious Case of Neural Text Degeneration, ICLR 2024.
> > [3]. Meister et al, Locally Typical Sampling, ACL 2023.
> > [4]. Fu et al, Beyond Reproducibility: Token Probabilities Expose Large Language Model Nondeterminism, arxiv 2026.

---

### Author Response · Authors · 2026-07-16
**General Response To Reviewers**

We thank all reviewers for providing constructive feedback on the paper which is of utmost value to our work. Below we provide a list of changes implemented in the manuscript. These changes can be found in $\color{blue}{\text{blue}}$. We further address your individual concerns in the responses below.
* [reviewer Pt7T] Additional motivation of confabulations and their effects on downstream commercial goals
* [reviewer Pt7T] Addition of effects of hallucinations and confabulations in the Broader Impact Statement
* [reviewer Pt7T, Mun2, k8oV] Additional experiments and ablations comparing the checklist to prompting and temperature scaling methods in Table 5, Figure 12 and Figure 13
* [reviewer Mun2, k8oV] Addition of mathematical definitions for sum of activations above mean, top-k operation, SAE objective and clarification of mathematical notation
* [reviewer Mun2] Addition of steps on computation of entropy and skewness in Appendix B
* [reviewer Mun2] Addition of discussion justifying the usage of PCA and SAE for dimensionality matching in Section 5
* [reviewer Mun2, k8oV] Additional ablations in Figure 24 comparing the role of prompt prefixes with and without finetuning
* [reviewer Mun2] Additional experiments validating negative correlation between layer activity and confidence ratings in Figure 25
* [reviewer Mun2] Additional experiments for randomly selected thresholds with varying finetuning effort and steering strength in Table 6
* [reviewer Mun2] Restructuring of related work in Section 2
* [reviewer Pt7T, Mun2] Fixing of minor grammatical errors